**TOOLS**

# High-content phenotyping reveals Golgi dynamics and their role in cell cycle regulation

Xun Cao[1]*, Yiming Peng[1]*, Mengyuan Yang[1]*, Mengling Gan[1], Di Zhang[1], Shiyue Zhou[1], and Daisuke Takao[1,2]

**Recent advances in quantitative bioimage analysis have enabled detailed analyses of cellular and subcellular morphological features, enhancing our understanding of cellular functions. Here, we introduce an image-based phenotyping pipeline designed for the comprehensive analysis of dynamic organelle morphology, particularly the Golgi apparatus and cilia, during cell cycle progression. Our approach emphasizes interpretable feature extraction, enabling detection of both prominent and subtle morphological changes. By using well-characterized morphological dynamics of intracellular structures as benchmarks, we demonstrated that our method can reliably detect established phenotypic changes and serves as a valid tool for quantitative profiling. Further investigation of the G0/G1 transition revealed an unexplored link between Golgi dynamics and ciliary disassembly. Specifically, inhibition of the G0/G1 transition correlated with ciliary persistence and unique Golgi dispersion, involving Aurora kinase A (AURKA). Our results thus indicate an association of Golgi morphology with cell cycle reentry and ciliary dynamics, underscoring the value of our profiling method in studying cellular regulation in health and disease.**

## Introduction

Cells exhibit remarkable structural plasticity, dynamically altering their shape and organelle configurations in response to both internal and external signals (Prosser and Pelletier, 2017; Scepanovic and Fernandez-Gonzalez, 2024). These morphological changes, particularly organelle interactions, are essential for understanding cellular function and regulation (Kwak et al., 2020; Vallese et al., 2020; Voeltz et al., 2024). To fully understand these processes, it is essential to employ a comprehensive approach that analyzes the dynamic morphological changes and interactions of multiple cellular structures and organelles. Additionally, instead of focusing on overly specialized features, it is beneficial to profile generalizable, quantifiable characteristics, such as the variance in the intracellular distribution of organelles.

Recent advances in computational image analysis have revolutionized cellular morphology studies, shifting from error-prone manual methods to high-throughput, precise quantification (Moen et al., 2019; Chai et al., 2024). Deep learning algorithms, such as cellpose (Stringer et al., 2021; Pachitariu and Stringer, 2022), Usiigaci (Tsai et al., 2019), and Deepcell (Bannon et al., 2021; Greenwald et al., 2022), have facilitated the extraction of individual cell boundaries from crowded populations, enabling large-scale data analysis with minimal manual intervention. Advances in cellular morphology analysis now enable the

quantification of features, such as size, shape, and texture, providing insights into cellular states (Govek et al., 2023; Kołodziej et al., 2023; Laan et al., 2023; Mysior and Simpson, 2024; Berg et al., 2019; Sommer et al., 2017; Stossi et al., 2024). Integrating these data with high-throughput omics technologies has expanded the scope of systems biology and genome-wide phenotypic screening (D'Ambrosio and Vale, 2010; Yan et al., 2021; Funk et al., 2022). Deep learning–based approaches further aid in detecting subtle morphological changes relevant to cellular states (Nagao et al., 2020). However, a key challenge remains in optimizing these methods to focus on interpretable features directly correlated with cellular functions. Rather than simply classifying cells, acquiring large-scale datasets designed to analyze defined cellular functions is an essential goal for addressing this issue and advancing the field.

Among cellular organelles, the Golgi apparatus plays a pivotal role in cellular homeostasis, intracellular trafficking regulation, and modification of proteins and lipids (Li et al., 2019; Mohan et al., 2023). Throughout the cell cycle, the Golgi undergoes regulated structural changes, particularly evident during mitosis when it disassembles into a dispersed "haze" pattern to ensure even distribution to daughter cells (Colanzi et al., 2003). In late G2, Golgi ribbon structures dissociate into isolated stacks in a process called "unlinking," which is thought to serve as a

[1]College of Animal Sciences and Technology and College of Veterinary Medicine, Huazhong Agricultural University, Wuhan, China; [2]Hubei Hongshan Labolatory, Wuhan, China.

*X. Cao, Y. Peng, and M. Yang contributed equally to this paper. Correspondence to Daisuke Takao: dtakao@mail.hzau.edu.cn

Y. Peng's current affiliation is Department of Biosciences, Durham University, Durham, UK.

checkpoint for the G2/M transition (Sütterlin et al., 2002; Hidalgo Carcedo et al., 2004; Yoshimura et al., 2005; Colanzi et al., 2007). Although Golgi unlinking closely associates with proper cell cycle progression, the precise functional relationship between Golgi morphology and cell cycle regulation remains unclear.

Most studies have focused on the dramatic Golgi changes during late G2 and mitosis, but relatively little attention has been given to its behavior during the G0/G1 transition. Of particular interest is the relationship between the Golgi and primary cilia (hereafter termed cilia). Cilia are hair-like projections that act as cell's antennae to receive external signals. During the quiescent G0 phase, cells assemble cilia, which disassemble upon cell cycle reentry (Doornbos and Roepman, 2021; Mill et al., 2023). Although ciliary dynamics integrate closely with cell cycle regulation, their exact role in this process is not fully understood (Izawa et al., 2015; Fabbri et al., 2019; Kasahara and Inagaki, 2021). The Golgi apparatus mediates material transport to cilia and frequently localizes adjacent to cilia, highlighting a morphological relationship critical for ciliogenesis and cilia maintenance (Masson and El Ghouzzi, 2022; Stevenson, 2023; Jin et al., 2022). Understanding Golgi dynamics during the G0/G1 transition, therefore, is essential, as it could provide insights into ciliary disassembly and the regulation of cell cycle reentry, which has yet to be fully explored.

In this study, we employed a high-content phenotyping approach to generate extensive datasets, characterizing cellular and subcellular morphological dynamics, and correlating them with cell cycle progression. Specifically, we analyzed Golgi morphology and its dynamics during the G0/G1 transition, a topic insufficiently explored in previous research. We emphasize the comprehensive, quantitative nature of our datasets, focusing on interpretable features that capture subtle Golgi changes, mitotic spindle defects, and cell cycle–dependent phenotypes. Importantly, our findings highlight the potential biological significance of the Golgi-cilia axis in cell cycle regulation, contributing valuable insights into cellular processes and enriching the field of cell morphology and phenomics.

## Results

### An image-based single-cell phenotyping pipeline integrating quantitative extraction of subcellular structures

We first established a pipeline to obtain morphological features of individual cells and generate cell phenotype profiles (Fig. 1 A). In this approach, cell samples co-stained for intracellular structures by immunofluorescence (IF) are observed under a microscope, and the resulting images are subjected to a deep learning–based segmentation algorithm, cellpose (Stringer et al., 2021; Pachitariu and Stringer, 2022), to identify individual cells (Fig. S1 A). After extracting morphological features of the cells and their internal structures, we perform multivariate analysis to create detailed phenotype profiles. This enables detection of clusters within the population that differ by subtle morphological features. Rather than simply clustering cells, our focus on interpretable features aims to advance the understanding of cell biological processes. We also developed the pipeline in Python in

Jupyter notebook format, integrating image processing and analysis on a seamless platform.

Next, we sought to quantitatively extract Golgi morphological features, since the Golgi apparatus undergoes dynamic changes that play significant functional roles in various processes, including cell cycle progression. Several quantitative indicators have been proposed, such as counting Golgi fragments (Mascanzoni et al., 2024), measuring Golgi volume (Frye et al., 2023) or area (Wortzel et al., 2017), analyzing sub-Golgi protein localization (Tie et al., 2016), or spatial features of the three-dimensional Golgi morphology and its association to the centrosome (Frye et al., 2023). Despite these proposals for quantitative analysis, most studies still rely on conventional qualitative classification. Therefore, a robust and accessible method is needed for more objective, efficient, and flexible quantitative analysis, suitable for a wide range of experimental settings. Moreover, capturing subtle morphological changes requires a comprehensive approach combining multiple parameters. As one strategy, we represented the Golgi as a set of discrete points by using the Trackpy algorithm (Allan et al., 2023) to detect fluorescence intensity peaks (puncta), and then characterized the Golgi based on the spatial distribution of these peaks. For example, a densely packed Golgi yields fewer fluorescence peaks with smaller coordinate variance, while a dispersed Golgi produces more peaks and greater variance (Fig. 1 B and Fig. S1 B). Because a packed Golgi often resides near the nuclear periphery, measuring the distance between the Golgi's center of mass and that of the nucleus also provides a useful parameter (Fig. 1 C and Fig. S1 B). As an alternative approach, we used skeletonization, a method that represents objects as lines by removing thickness, to abstract the Golgi morphology. Fragmented Golgi structures tend to yield numerous short line objects, whereas more compact Golgi structures give rise to fewer, longer "tubular" objects (Fig. S1 C). The Golgi-related features described in this study represent apparent morphology observed in relatively low-resolution images optimized for throughput. Rather than providing definitive structural details, these features offer a basis for subsequent higher-resolution analyses and complementary molecular or cell biological investigations.

While we focused on the Golgi as an example, our method of extracting cell morphological features can be also applied to other subcellular structures. For example, the number of mitotic spindle poles can be determined via fluorescence peak detection, and cilia morphology can be described through skeletonization. The features used in this study are listed in Table S1. Because our pipeline relies entirely on quantitative indicators, it offers comprehensive, unbiased analyses without qualitative judgments. As we demonstrate below, this has enabled us to efficiently obtain the large datasets required to characterize cell biological processes.

### The image-based cell phenotype profiling identified features related to morphological defects in mitotic cells

To demonstrate the capabilities of our cell phenotype profiling, we first applied it to an analysis of mitotic spindle structure, which undergoes distinct morphological changes. In this experiment, we treated synchronized HeLa cells with DMSO

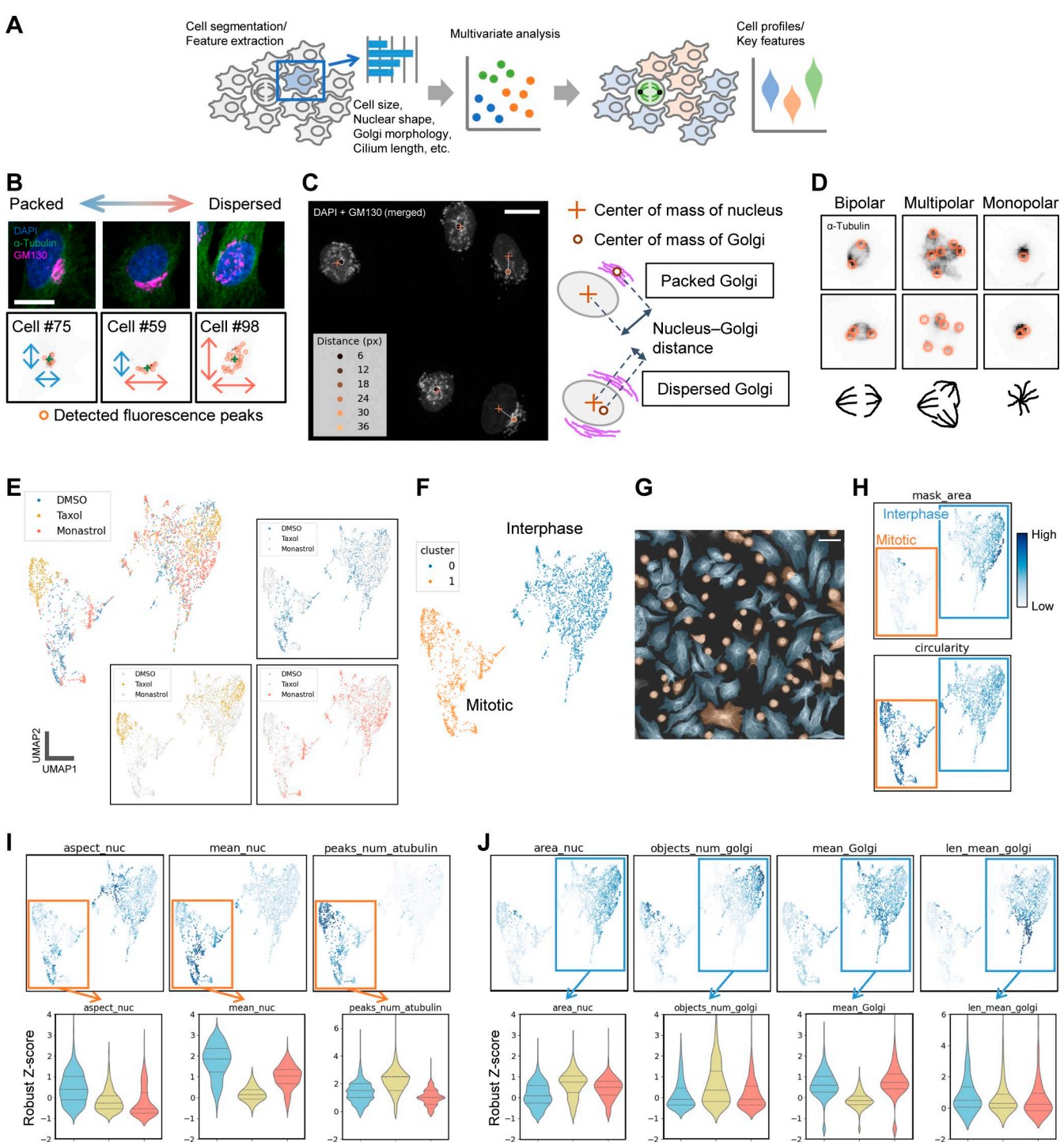

Figure 1. **Image-based single-cell phenotype profiling and analysis of the effects of microtubule inhibitors on mitotic cell profiles. (A)** Schematic of the single-cell phenotype profiling pipeline. **(B and C)** Representative methods for characterizing Golgi morphology. Golgi morphology was quantified by measuring fluorescence peaks (puncta) in GM130-stained images. (B) Peak distribution and (C) the nucleus–Golgi distance were assessed. Scale bars, 20 µm. **(D)** Characterization of mitotic spindle structure via fluorescence peak extraction. Fluorescence peaks from α-tubulin images were used to distinguish bipolar, multipolar, or monopolar spindle structures. **(E)** UMAP plot of cell phenotype profiles. Three experimental conditions are shown in color-coded form (top left) or individually highlighted for clarity. Each point represents a single-cell profile. **(F and G)** DBSCAN clustering results. The two identified clusters are color-coded in F, and these colors are overlaid on an original image of DMSO-treated cells (G). Based on this mapping and subsequent analysis, clusters 0 and 1 correspond approximately to interphase and mitotic cells, respectively. Scale bar, 50 µm. **(H)** Projection of key features distinguishing interphase and mitotic cells onto the UMAP plot. Here, "mask_area" and "circularity" represent the area and circularity of the cell mask. **(I)** Features reflecting the effects of drug treatment on mitotic cells. Violin plots (with dashed lines indicating the median and quartiles) display aspect_nuc (aspect ratio of ellipse-fitted chromosomes), mean_nuc (mean DAPI fluorescence intensity), and peaks_num_atubulin (number of fluorescence peaks from α-tubulin staining). **(J)** Features showing the impact of drug treatment on interphase cells. Measured parameters include area_nuc (nuclear size), objects_num_golgi and len_mean_golgi (number and mean length of skeletonized Golgi structures), and mean_Golgi (mean fluorescence intensity of GM130 staining).

(control), taxol, or monastrol for 6 h (Fig. S1 D) and stained for DAPI (DNA), α-tubulin (microtubules), and GM130 (Golgi marker) after fixation (Fig. S1, D and E). As expected, most mitotic cells in the control group formed normal bipolar spindles, whereas many multipolar and monopolar spindles were observed under taxol and monastrol treatment, respectively (Fig. S1 E). We confirmed that spindle poles could be reasonably detected by extracting fluorescence intensity peaks (bright spots or puncta) in α-tubulin images (Fig. 1 D). Although α-tubulin is not a specific spindle pole marker, and thus the number of these bright spots does not necessarily match the exact number of spindle poles, it serves as a useful index for characterizing spindle structure. In addition, because the microtubule network can provide information beyond spindle pole count, α-tubulin can be a valuable marker for high-content analyses. All features used in our analysis are listed in Fig. S1 F and Table S1.

Our cell phenotype profiling method provides two practical advantages: ease of use and computational efficiency. To promote an open environment in high-content, high-throughput imaging, we designed the method to be accessible even to small research groups with limited resources. Using a standard office computer (Core i5-12400, 16 GB RAM), it took 46.3 s to obtain all single-cell profiles from the image shown in Fig. S1 A. Of this, the segmentation process required 20.6 s, which could be further accelerated with a GPU; for example, the same segmentation took only 2.3 s on Google Colab with a T4 GPU. This enables the processing of more than 100 images per hour, corresponding to over 5,000 single-cell profiles if each image contains 50 cells. Although minimal coding skills are required, the code is simple and can be easily modified by users. As noted above, another advantage of our approach is that it extracts abstract morphological features rather than focusing on a specific structure, thereby providing versatility with respect to the analytical objective. The code is available in a public repository (see Data availability).

We next performed a detailed analysis of cell profiles based on these extracted features. Two-dimensional UMAP plots clearly separated cells based on drug treatment conditions (Fig. 1 E). In parallel, DBSCAN classified the cell population into two main clusters (clusters 0 and 1), regardless of drug treatment (Fig. 1 F). Mapping these cluster labels back to the original images revealed that interphase and mitotic cells were predominantly in clusters 0 and 1, respectively (Fig. 1, F and G and dataset 1A; all datasets referenced in this paper are deposited in Dryad as noted in the Data availability section). To identify key features distinguishing interphase and mitotic cells, we projected each feature's value onto the UMAP plots (dataset 1B). Dataset 1C shows a heat map of the mean value of each feature for each cluster and drug condition. The full distribution of all features is shown in dataset 2. Notably, differences in overall cell size (mask_area) and circularity were particularly prominent (Fig. 1 H), consistent with the known rounding and reduced size of metaphase cells. Thus, as expected, we first distinguished mitotic cells from interphase cells based on clear morphological differences.

We then investigated whether our cell phenotype profiling could detect features that define spindle assembly defects more precisely. Focusing on mitotic cells only (cluster 1), both the aspect ratio of chromosomes in the images (aspect_nuc) and mean DAPI fluorescence (mean_nuc) were markedly reduced in taxol- and monastrol-treated cells compared with control cells (Fig. 1 I). This aligns with the observation that in normal metaphase, chromosomes are tightly packed and elongated at the spindle center, whereas in monopolar or multipolar spindles, they are more dispersed and irregular. Consistent with Fig. 1 D, the number of α-tubulin fluorescence peaks (peaks_num_atubulin) reflected spindle pole numbers: two peaks were typically found in the control, whereas taxol treatment tended to produce more and monastrol fewer (Fig. 1 I). Note that the vertical axis of the violin plots in Fig. 1 I represents standardized robust z-scores, not absolute values. These results demonstrate that our cell phenotype profiling not only distinguishes mitotic cells from interphase cells but also identifies subtle variations in spindle structure and related phenotypic features.

### Beyond major mitotic defects: Our approach detected subtle changes in Golgi morphology in interphase cells

In addition to the prominent differences in mitotic spindle structure, our cell phenotype profiling also captured subtle morphological changes in interphase cells under drug treatment. These known changes were used as benchmarks to further evaluate the performance of our analytical approach. Focusing on interphase cell profiles, nuclear size (area_nuc) tended to increase in both the taxol- and monastrol-treated groups compared with the control group (Fig. 1 J), likely reflecting the effect of inhibiting microtubule dynamics on nuclear structure (Tariq et al., 2017). In taxol-treated cells, the number of detected Golgi fragments (objects_num_golgi) increased, whereas mean fluorescence intensity (mean_Golgi) and mean fragment length (len_mean_golgi) decreased slightly (Fig. 1 J), indicating fragmentation and dispersion, as previously reported (Wehland et al., 1983; Hoshino et al., 1997). A similar trend, though milder, was observed in monastrol-treated cells (Fig. 1 J).

Overall, these results demonstrate the sensitivity of our method for detecting subtle alterations in specific morphological features, highlighting its utility in quantitative characterization of cellular and subcellular structures, as well as their regulation. In particular, the ability to objectively characterize organelle morphology, such as Golgi structure, motivates further investigation into the mechanisms regulating subcellular organization and cell cycle progression.

### Golgi morphology during late G2 quantitatively analyzed to detect subtle changes relevant to G2/M transition

The morphology of the Golgi changes dynamically from late G2 to early G1 (Fig. 2 A), which may be involved in cell cycle checkpoint mechanisms. In addition, upon exiting the cell cycle and entering G0, cells form cilia using materials transported from the adjacent Golgi, and these cilia are disassembled upon reentry into the cell cycle (Fig. 2 A). However, the role of Golgi morphology in ciliary disassembly has remained poorly characterized. Therefore, detailed analysis of Golgi morphology dynamics during interphase may shed light on previously unknown cellular mechanisms.

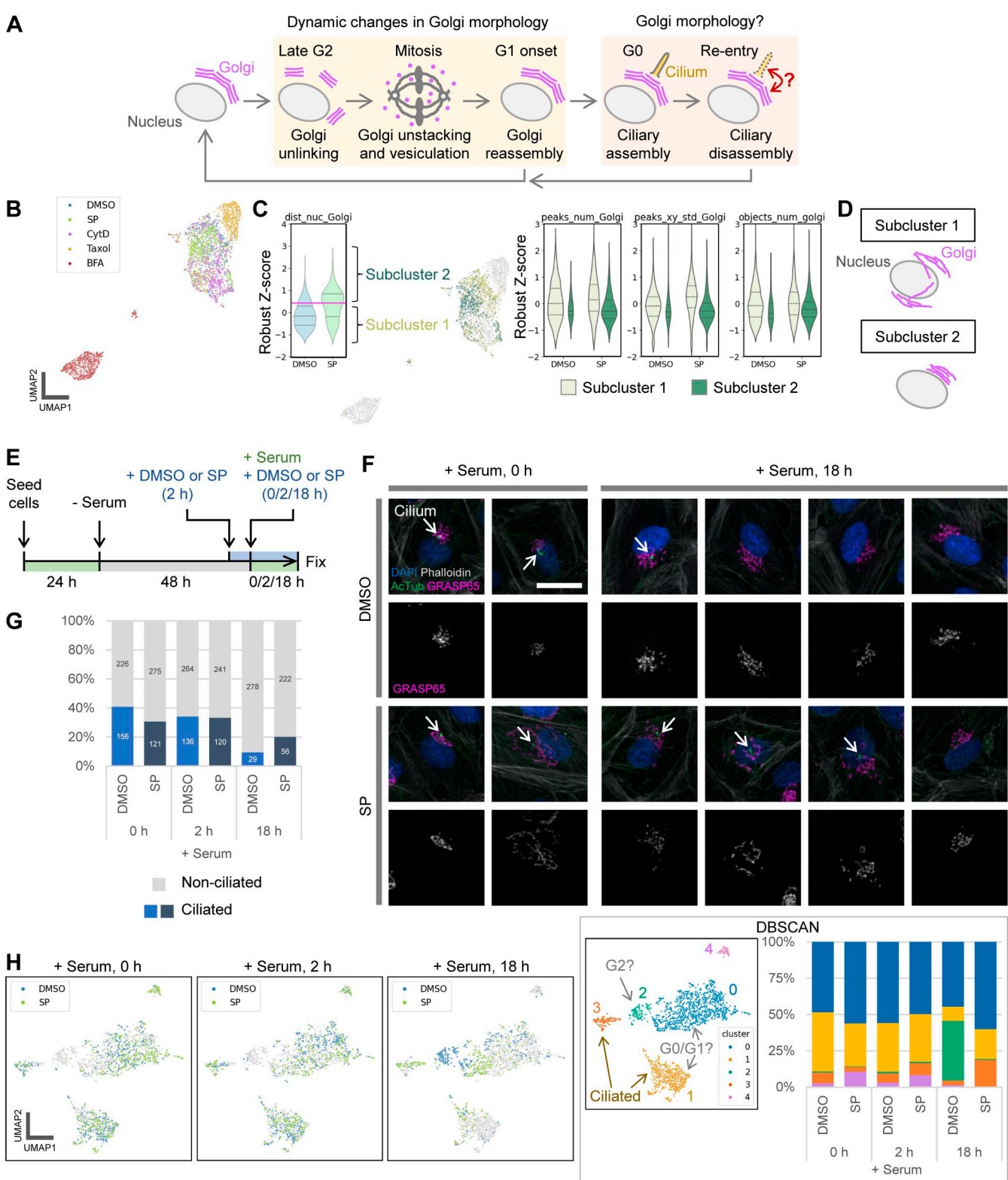

Figure 2. **Analysis of changes in cell phenotype profiles, including Golgi morphology, during late G2 and G0/G1 transition. (A)** Schematic illustrating the morphological dynamics of the Golgi and cilia throughout the cell cycle. This figure primarily focuses on Golgi dynamics in late G2, while subsequent figures address the G0/G1 transition. **(B–D)** Morphological feature analysis of HeLa cells in late G2. **(B)** UMAP plots of cell phenotype profiles in late G2. All experimental conditions are shown in a composite color scheme, with each condition also highlighted individually in Fig. S2 D. Representative images are shown in Fig. S2 B, and the features used in this analysis are detailed in Fig. S2 C. **(C)** Analysis of Golgi-based subclusters in the control and SP-treated groups. Data were extracted from the dist_nuc_Golgi violin plot in Fig. S2 E and divided into subclusters 1 and 2 using the SP group's median (indicated by the magenta line in the leftmost violin plot). Distributions of three representative features, i.e., peaks_num_Golgi, peaks_xy_std_Golgi, and objects_num_golgi (number of Golgi fragments detected after skeletonization), are shown for each experimental condition and subcluster. **(D)** Schematic representation of typical Golgi

morphologies for each subcluster, based on the analysis. **(E–H)** ARPE-19 cells were serum starved to arrest in G0 and induce ciliogenesis. Morphological changes were then analyzed after serum re-addition. **(E)** Time course of serum starvation and serum re-addition. DMSO (control) or SP was added 2 h before serum re-addition, and cells were then cultured with serum and the indicated drugs for the specified durations before fixation. For the 0-h time point, cells were fixed immediately without serum re-addition. **(F)** Cropped images of representative cells at 0 and 18 h after serum re-addition. Arrows indicate cilia. AcTub, acetylated tubulin. Scale bar, 20 μm. **(G)** Percentage of ciliated cells at each time point after serum re-addition. Numbers in the bar graphs represent absolute cell counts for each category. Cilia were detected by skeletonizing the acetylated tubulin signal. **(H)** UMAP visualization using features detailed in Fig. S3 B. Time points for the DMSO (control) and SP-treated groups are highlighted for clarity. DBSCAN clustering results and the proportion of cells in each cluster are also shown in the box.

---

Because we detected subtle changes in Golgi morphology during interphase, we next focused on late G2, immediately before mitosis, to analyze changes in Golgi morphology associated with cell cycle progression (Fig. 2 A). Although unlinking of Golgi stacks is crucial for the G2/M transition (Ayala and Colanzi, 2022; Iannitti et al., 2025), its morphological changes are relatively mild, and more detailed morphological analysis remains challenging. We, therefore, analyzed cellular and subcellular structures, including the Golgi, during late G2 under several established experimental conditions to further validate the sensitivity of our cell phenotype profiling pipeline. HeLa cells synchronized by a double thymidine block were treated with different inhibitors, and then fixed and immunostained (Fig. S2 A). Golgi unlinking has been shown to involve the phosphorylation of the Golgi reassembly and stacking protein GRASP65, mediated by JNK 2 (JNK2). Inhibition of GRASP65 functions via JNK2 inhibitors, such as SP600125 (SP), disrupts Golgi unlinking, blocking mitotic entry, and leading to defective spindle assembly (Cervigni et al., 2015; Mascanzoni et al., 2024). Along with SP, we used the common inhibitors cytochalasin D (CytD; actin polymerization inhibitor), taxol (microtubule depolymerization inhibitor), and brefeldin A (BFA; intracellular transport inhibitor) for comparison. In addition to the Golgi apparatus (GM130), we used features related to the actin cytoskeleton (phalloidin), nucleus/DNA (DAPI), and the proliferation-related nuclear protein Ki-67 (Fig. S2, B and C). Analysis of these features separated the cell profiles into distinct clusters based on drug treatment (Fig. 2 B and Fig. S2 D). BFA caused a dramatic effect, forming a cluster entirely separate from the other treatments. The remaining four conditions, including the control, formed partially overlapping but mostly distinct clusters (Fig. 2 B and Fig. S2 D), consistent with qualitative observations (Fig. S2 B).

Next, we examined which features changed characteristically under each condition (dataset 3). The results revealed features that reflect several distinct Golgi morphologies. In the control group during late G2, the Golgi appeared relatively dispersed, possibly reflecting unlinking, whereas SP treatment frequently yielded "packed" Golgi (Fig. S2 B). These changes were relatively subtle and some SP-treated cells resembled control cells, consistent with previous reports (Cervigni et al., 2015; Mascanzoni et al., 2024; Barretta et al., 2016). Correspondingly, the control and SP-treated populations overlapped on the phenotype map (Fig. 2 B and Fig. S2 D). Nevertheless, the distance between the Golgi and nucleus centers of mass (dist_nuc_Golgi), an indicator of Golgi dispersion (Fig. 1 C), tended to increase under SP treatment (Fig. S2 E). The SP-treated group also showed a bimodal distribution for this nucleus–Golgi distance, suggesting

subpopulations with either similar or more pronounced Golgi packing compared with the control.

Interestingly, Ki-67 features also changed strikingly under SP treatment. In particular, the coefficient of variation (CV) of Ki-67 fluorescence (cv_ki67) was lower than in the control (Fig. S2 F). As shown in Fig. S2 B, Ki-67 staining became more homogeneous in SP-treated cells, with fewer bright intranuclear objects or nonuniform textures. Unlike the Golgi features, the Ki-67 fluorescence CV had a unimodal distribution, indicating that most SP-treated cells showed this uniform staining pattern.

Other treatments produced distinct profiles that were largely consistent with known morphological responses, thereby providing additional validation for the accuracy of our cell phenotype profiling. Taxol induced well-known Golgi fragmentation (Wehland et al., 1983; Hoshino et al., 1997) (Fig. S2 B), reflected by a higher number of GM130/Golgi fluorescence peaks (peaks_num_Golgi) and greater spatial variance (peaks_xy_std_Golgi) (Fig. S2 E). BFA led to the most dramatic changes, causing the Golgi to lose its rigid structure and adopt a "hazy" appearance reminiscent of, but slightly different from mitotic Golgi (Fig. S2 B). This was detected as an increased area (area_Golgi) and decreased mean fluorescence intensity (mean_Golgi) of GM130/Golgi, consistent with Golgi disorganization (Fig. S2 E). In contrast, CytD caused mild Golgi packing, indicated by slightly increased GM130/Golgi fluorescence (mean_Golgi, Fig. S2 E) and reduced Golgi fragment numbers (objects_num_golgi, Fig. S2 F), but more global effects on cell shape, including reduced cell (mask_area) and nuclear size (area_nuc) and increased variation in phalloidin/actin signals (cv_actin) (Fig. S2 F). Taxol and BFA had no noticeable effects on cell or nuclear morphology, instead selectively altering the Golgi among the features analyzed (Fig. S2, E and F). Thus, each inhibitor, including SP, differentially affected Golgi and other subcellular structures in a manner consistent with previous studies, and our profiling approach captured the key features underlying these morphological patterns.

Because the nucleus–Golgi distance suggested two subclusters within the SP-treated group, we further analyzed these subpopulations. Focusing only on the control and SP-treated groups for simplicity, we divided these cells into subclusters 1 and 2 by the median nucleus–Golgi distance (dist_nuc_Golgi) of the SP-treated group (Fig. 2 C). Most control cells fell into subcluster 1, suggesting it represented "normal" cells, while subcluster 2 comprised cells with more pronounced Golgi changes under SP treatment. Comparison of the features of each subcluster (dataset 4) showed that subcluster 2 had lower numbers and spatial variance of GM130/Golgi fluorescence peaks (peaks_num_Golgi and peaks_xy_std_Golgi) and fewer Golgi fragments

(objects_num_golgi), collectively indicating more packed Golgi morphology (Fig. 2, C and D). These results demonstrate that effects of SP on Golgi morphology vary among cells, and our cell phenotype profiling can detect subpopulations with distinct sensitivities.

### The inhibitor of G2/M Golgi unlinking affects Golgi morphology and cilia disassembly at the G0/G1 transition

Next, we aimed to gain mechanistic insights into how Golgi morphology relates to other subcellular structures by focusing on the functional connection between the Golgi and cilia during cell cycle reentry from quiescence (G0 phase; Fig. 2 A). Given that (1) Golgi morphological changes can function as a checkpoint regulating the G2/M transition (Ayala and Colanzi, 2022; Iannitti et al., 2025), (2) the Golgi supplies materials required for cilia formation and maintenance (Masson and El Ghouzzi, 2022; Stevenson, 2023; Jin et al., 2022), and (3) cilia are known to participate in cell cycle progression (Izawa et al., 2015; Fabbri et al., 2019; Kasahara and Inagaki, 2021), analyzing largely unexplored Golgi morphology during the G0/G1 transition could provide clues about novel Golgi functions in cell cycle regulation.

To investigate cell cycle reentry from quiescence, we arrested ARPE-19 cells in G0 phase by serum starvation, then observed changes in cell phenotype profiles after serum re-addition (Fig. 2 E). DMSO (control) or SP was added 2 h before serum re-addition and maintained throughout. A relatively low cell seeding density ($0.5 \times 10^5$ cells per well in a 12-well plate) was used in this experiment to avoid cell polarization, which may hinder cell cycle reentry. Under our conditions, about 40% of cells in the control group were ciliated after 48 h of serum starvation (Fig. 2, F and G; and Fig. S3 A; DMSO, 0 h), which is slightly lower than but still largely consistent with previously reported frequencies ranging from 43% to ∼70% (Wang and Brautigan, 2008; Gómez et al., 2022). Following serum re-addition, the proportion of ciliated cells decreased moderately at 2 h and more substantially at 18 h (late G2) (Fig. 2, F and G; and Fig. S3 A). With SP treatment, however, slightly fewer cells were initially ciliated, yet a relatively large fraction retained cilia at both 2 and 18 h compared with the control (Fig 2, F and G). These findings suggest that SP treatment hinders ciliary disassembly upon cell cycle reentry or impedes the G0/G1 transition itself.

To assess the involvement of Golgi structure and function in this process, we analyzed time course cell phenotype profiles based on DAPI (DNA), phalloidin (actin filaments), acetylated tubulin (cilia), and GRASP65 (Golgi) staining (Fig. 2 F and Fig. S3 B). Over the course of serum re-addition, both the control and SP-treated groups exhibited changes in cell phenotype profiles, but with notable differences (Fig. 2 H). Immediately before (0 h) and shortly after (2 h) serum re-addition, SP-treated cells diverged from control cells, particularly among non-ciliated cells, partly separating within cluster 0 (non-ciliated) and giving rise to cluster 4, which was specific to SP (Fig. 2 H). By 18 h after serum re-addition, these differences became more pronounced: most control cells lacked cilia and formed cluster 2 (non-ciliated), typical of this stage, whereas SP-treated cells remained in cluster 1 (ciliated), cluster 3 (partially ciliated), and cluster 0 (non-ciliated) (Fig. 2 H). In cluster 2, cells showed significantly

increased mean DAPI fluorescence intensity (mean_nuc), Golgi area (area_Golgi), and number of GRASP65/Golgi fluorescence peaks (peaks_num_Golgi), indicative of Golgi dispersion and entry into G2 (dataset 5). The increase in the mean fluorescence intensity of DAPI suggests an increase in DNA content, i.e., the cells have reentered the cell cycle, passed through the S phase, and entered the G2 phase. Cluster 3 shared similarities with cluster 2, including larger cell size (mask_area) and broadly distributed Golgi, but it differed by lacking the rise in mean DAPI fluorescence intensity (mean_nuc) and showing a slightly greater nucleus–Golgi distance (dist_nuc_Golgi); it also had a bimodal distribution of cilia numbers (objects_num_cilia) (dataset 5). In contrast, cluster 4 comprised cells with smaller cell size (mask_area), narrower Golgi area (area_Golgi), and no cilia (dataset 5). Thus, SP treatment substantially affects intracellular structures, including the Golgi and cilia, at 18 h after serum re-addition, prompting us to focus on that time point for further analysis.

### Morphological changes and disruption of the G0/G1 transition are likely mediated by AURKA inhibition

To understand the mechanism underlying the disruption of intracellular structures at the G0/G1 transition induced by SP treatment, we examined which kinases are involved. SP is a broad-spectrum inhibitor that targets not only JNK2 but also other kinases such as JNK1 and Aurora kinase A (AURKA) (Bennett et al., 2001; Kim et al., 2010). To identify the specific kinases responsible for the disruption induced by SP, we depleted potential SP targets, including JNK1, JNK2, and AURKA, in ARPE-19 cells using RNAi and compared their phenotype profiles to those of SP-treated cells. After serum starvation and siRNA transfection, cells were incubated with DMSO (control) or SP for 18 h following serum re-addition, then analyzed (Fig. 3 A).

Before detailed cell phenotype profiling, we measured cell cycle progression by assessing EdU incorporation and Ki-67 content (mean_EdU and mean_ki67, respectively), both of which can reflect cell cycle reentry (Fig. 3 B and Fig. S3 D). The control group (siControl) exhibited higher fluorescence intensities for both markers, suggesting that many cells were in G2, whereas SP-treated cells had lower values, indicating that most were arrested in G0 (Fig. 3 B). The siJNK2 and siAURKA groups resembled the SP-treated group, showing low EdU and Ki-67 intensities, while the siJNK1 group resembled the control (Fig. 3 B). The frequency of ciliated cells reflected similar trends, with siJNK1-treated cells showing a slightly higher percentage than control cells (Fig. 3 B), supporting the conclusion that SP treatment, via inhibition of JNK2 and/or AURKA, likely caused G0 arrest in most cells by 18 h after serum re-addition.

To investigate Golgi morphology and its involvement in the G0/G1 transition, we performed comprehensive cell phenotype profiling using features extracted from DAPI (DNA), phalloidin (actin filaments), acetylated tubulin (cilia), and GRASP65 (Golgi) (Fig. 3 C; and Fig. S3, C and E). Clustering analysis based on these features revealed two major groups (Fig. 3 D): non-ciliated and ciliated clusters, with the number of cilia (objects_num_cilia) serving as an indicator (dataset 6A). As observed previously (Fig. 2 H), the control and SP-treated groups showed relatively mild separation with some overlap (Fig. 2 D).

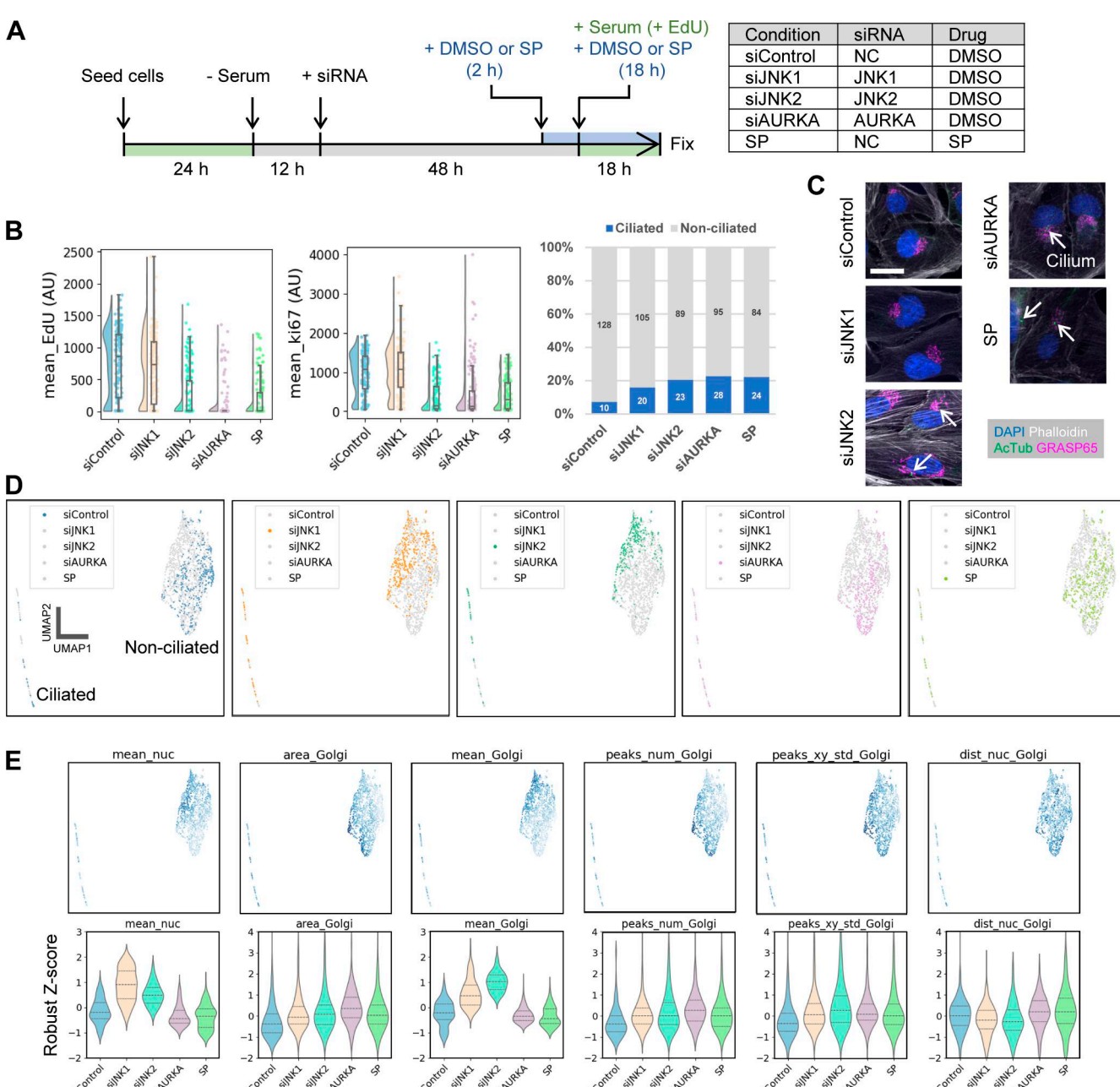

Figure 3. **Identification of factors and mechanisms underlying changes in cell phenotype profiles at the G0/G1 transition.** ARPE-19 cells were serum starved to arrest in G0 and induce ciliogenesis. Cell phenotype profiles were then analyzed 18 h after serum re-addition under various experimental conditions. **(A)** Time course of serum starvation and serum re-addition. Five experimental conditions were established by combining siRNA (with NC as the nontargeting control) and drug treatments (DMSO or SP), as detailed in the table on the right. EdU was added at the time of serum re-addition only in experiments designed to measure EdU incorporation. **(B)** Monitoring cell cycle progression and ciliation under each condition. Raincloud plots (Allen et al., 2021) show the distributions of the mean nuclear EdU (mean_EdU) and Ki-67 (mean_ki67) fluorescence intensities after background subtraction, alongside the percentage of ciliated cells (calculated as the ratio of cilia to nuclei). Representative images are shown in Fig. S3 D. **(C)** Representative cropped images of cells used in the cell phenotype profiling. Additional images are provided in Fig. S3 E. Arrows indicate cilia. AcTub, acetylated tubulin. Scale bar, 20 μm. **(D)** UMAP visualization using features from Fig. S3 C. Data for each experimental condition are highlighted separately for clarity. Two major clusters emerged, representing a predominantly non-ciliated group and a smaller ciliated group. **(E)** Analysis of notable features. UMAP and violin plots illustrate the distributions of several key parameters across experimental conditions: mean_nuc (mean DAPI/DNA fluorescence), area_Golgi (Golgi area), mean_Golgi (mean GRASP65/Golgi fluorescence intensity), peaks_num_Golgi (number of GRASP65 fluorescence peaks), peaks_xy_std_Golgi (SD of fluorescence peak coordinates), and dist_nuc_Golgi (distance between the nucleus and Golgi). Full data are available in dataset 6.

Among the RNAi groups, the profiles of AURKA-depleted cells aligned closely with those of SP-treated cells, while the siJNK1 and siJNK2 groups were more distinct (Fig. 3 D). For most features, siAURKA and SP-treated cells showed similar trends (dataset 6). For example, both groups contained cells with lower mean DAPI fluorescence (mean_nuc), indicating that these cells were likely arrested in G0 or early G1 (Fig. 3 E). Although not as pronounced, features related to Golgi morphology also showed a similar tendency in the siAURKA and SP-treated groups, i.e., increased area (area_Golgi), decreased mean fluorescence intensity (mean_Golgi), and higher numbers of fluorescence peaks (peaks_num_Golgi) and greater spatial variance (peaks_xy_std_Golgi), consistent with Golgi dispersion (Fig. 3 E). However, the nucleus–Golgi distance (dist_nuc_Golgi) did not decrease but rather increased slightly in these groups compared with the control (Fig. 3 E), which is not typical of dispersed Golgi (Fig. 1 C). This suggests that while the Golgi in these groups appeared dispersed, its center of mass remained in the perinuclear region, similar to packed Golgi, as shown in the schematic in Fig. S3 E. These findings suggest that the disruption in Golgi morphology observed in siAURKA and SP-treated cells is likely distinct from the typical Golgi unlinking observed during late G2 in the normal cell cycle.

The siJNK1 and siJNK2 groups exhibited somewhat similar profiles (Fig. 3 E and dataset 6) despite differences in cell cycle phase (Fig. 3 B). The Golgi in these cells tended to become dispersed (slight increases in area_Golgi, peaks_num_Golgi, and peaks_xy_std_Golgi), similar to siAURKA- and SP-treated cells, but the nucleus–Golgi distance was the same as or slightly smaller than that of the control, and the mean fluorescence intensity (mean_Golgi) was significantly increased (Fig. 3 E). The siJNK2 group also showed higher mean fluorescence intensity for phalloidin/actin (mean_actin) (dataset 6B). These profiles did not closely resemble those of the SP-treated group, suggesting that, despite its common use as a JNK2 inhibitor, the effect of SP in this context is primarily mediated by inhibition of AURKA rather than JNK1 or JNK2.

Given that the siJNK2-treated and siAURKA-treated groups showed similarities in some aspects, such as cell cycle–related phenotypes (Fig. 3 B), the phenotypic profiles of cells simultaneously depleted with these two genes were similarly analyzed to obtain more detailed information about these pathways (Fig. S4 A and dataset 7). As a result, the double knockdown phenotype was similar to that of siAURKA in terms of nucleus–Golgi distance (dist_nuc_Golgi), similar to that of siJNK2 in terms of the number of Golgi objects detected (objects_num_golgi), and similar to that of both siAURKA and siJNK2 in terms of the degree of Golgi dispersion (peaks_xy_std_Golgi) (Fig. S4 A). These results suggest that there is no synergistic effect or obvious cross talk between JNK2 and AURKA in this context.

To ensure the validity of the analysis, we then verified that the markers used were appropriate. Tubulin, which constitutes the axoneme, the skeleton of the cilia, is usually highly acetylated, and thus acetylated tubulin is frequently used as a cilia marker. However, it is known that tubulin deacetylation occurs prior to ciliary disassembly (Ran et al., 2015), so we examined whether acetylated tubulin is appropriate for use as a cilia

marker in the analysis of this study. To this end, we co-stained acetylated tubulin and IFT88 under each experimental condition. As expected, IFT88 was abundantly localized at the base and tip of the cilia, in addition to the shaft, and co-localized well with acetylated tubulin in all observed cases (Fig. S4 B). Therefore, we concluded that even if there were some changes in tubulin acetylation, acetylated tubulin would serve as an appropriate cilia marker in this analysis. Similarly, we co-stained two common Golgi marker proteins, GM130 and GRASP65, in each experimental condition. These two proteins well co-localized across all experimental conditions, and features related to Golgi morphology obtained independently from each cell showed a strong correlation (Fig. S4 C). Although more detailed morphological analysis will require the use of more markers that localize to different regions within the Golgi apparatus and/or electron microscopy, we conclude here that GRASP65 and GM130 can be used as interchangeable Golgi markers.

## Golgi and ciliary defects through G0/G1 transition are accompanied by reduced AURKA accumulation at the centrosome, but the spatial link between the Golgi and centrosome remains largely intact

AURKA is known to regulate ciliary disassembly and mitotic progression by accumulating at the basal body/centrosome (Pugacheva et al., 2007). To further demonstrate that AURKA is a major target of SP during the G0/G1 transition, we next quantified AURKA accumulation at the centrosome (including the basal body). To this end, we fixed cells according to the time course shown in Fig. 2 E and performed co-immunostaining for centrin, a centriole marker, and AURKA (Fig. 4 A and Fig. S5 A). Using our cell phenotype profiling pipeline, we extracted centrin signals to identify the position of the centrosome and automatically quantified AURKA intensity at the centrosome in individual cells. To perform cell segmentation, we utilized nonspecific background signals from the centrin staining. Although this approach slightly compromised segmentation accuracy for overall cell morphology, it was sufficient for reliable detection and quantification of centrosomal signals.

At 0 and 2 h after serum re-addition, all experimental conditions showed little to no detectable accumulation of AURKA at the centrosome. In contrast, by 18 h, a marked increase was observed in the control group (Fig. 4 A). At this time point, a subset of siJNK1- and siJNK2-transfected cells began to show a modest increase in AURKA accumulation, whereas the SP-treated group showed low accumulation levels comparable with those in the siAURKA group (Fig. 4 A). In summary, siJNK1, siJNK2, and SP treatment all impaired AURKA accumulation at the centrosome from the G0/G1 transition through the G2 phase, with SP treatment exhibiting a particularly strong inhibitory effect, comparable with that of siAURKA.

Next, to investigate the relationship between AURKA accumulation at the centrosome and the structural integrity between the Golgi apparatus and the centrosome, we analyzed their spatial relationship. By using the method in our cell phenotype profiling pipeline to calculate the distance between the Golgi and the nucleus, we measured the distances between the centers of

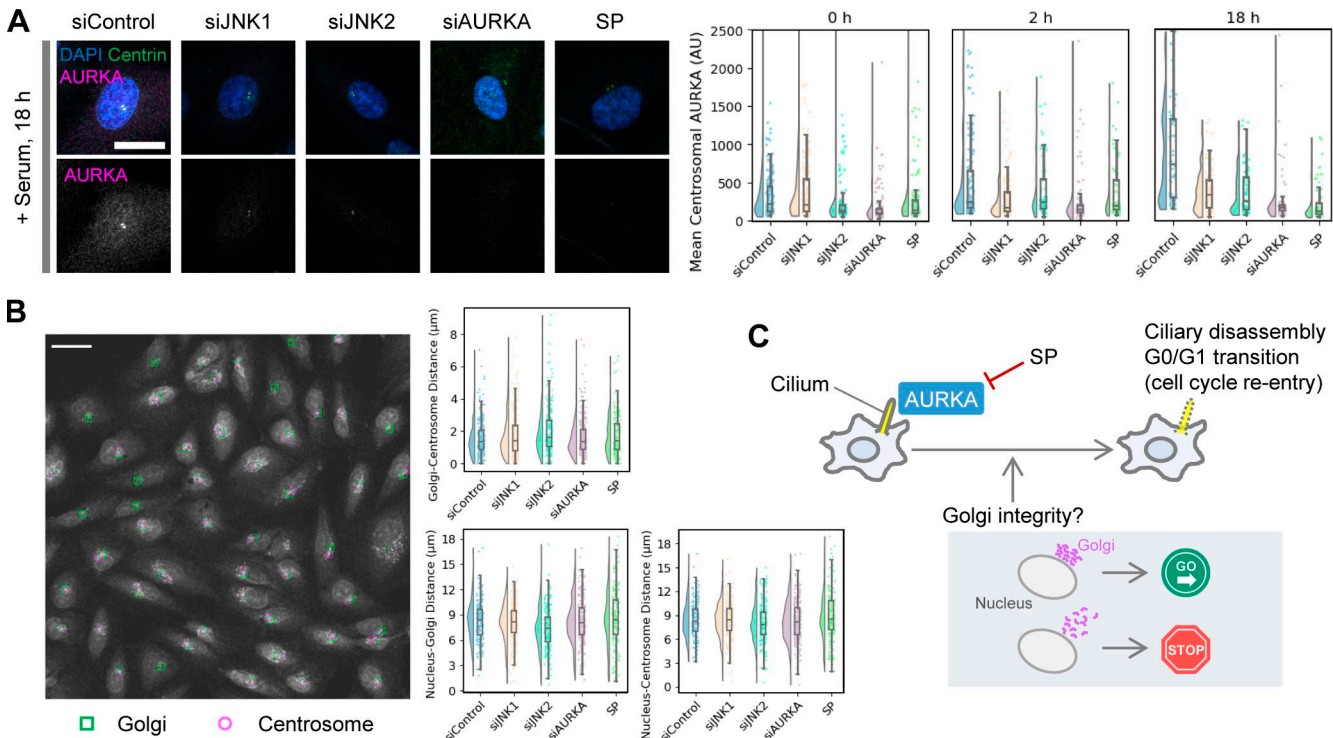

Figure 4. **Centrosomal AURKA accumulation and its functional implications in the G0/G1 transition. (A)** Analysis of AURKA accumulation levels at the centrosome along the time course in Fig. 2 E. Representative cropped images at 18 h after serum re-addition and raincloud plots of centrosomal AURKA levels at each time point are shown. Scale bar, 20 μm. Additional representative images are provided in Fig. S5 A. **(B)** Analysis of the relative positions of the Golgi and centrosome at 18 h after serum re-addition, following the time course in Fig. 3 A. The centers of mass of the nucleus, Golgi, and centrosome were determined from the DAPI, GM130, and γ-tubulin signals, respectively, and the distances between them were calculated. A representative image (siControl) with the centers of mass of the Golgi and centrosome overlaid on the merged image, along with raincloud plots of the measured distances, is shown. Scale bar, 50 μm. **(C)** Proposed model illustrating the mechanisms regulating the G0/G1 transition.

mass of the Golgi, centrosome, and nucleus in individual cells (Fig. 4 B). The results revealed that the tendency for the centrosome to be located near the Golgi was not substantially affected by these experimental conditions, and the Golgi–centrosome spatial relationship remained largely stable (Fig. 4 B). Although siAURKA and SP treatments induced Golgi fragmentation (Fig. S3 E), no major structural disruption was observed that would significantly disturb the positional relationship between the Golgi and centrosome.

## Discussion

In this study, we introduced an image-based single-cell phenotype profiling technique to analyze dynamic morphological changes of cellular and subcellular structures. We particularly focused on key organelles, such as the Golgi apparatus and cilia, and generated comprehensive datasets for detailed analyses of cellular functions across different stages of the cell cycle. This approach can also be extended to other organelles exhibiting dynamic behaviors, such as mitochondrial fission–fusion cycles (Wang et al., 2019; Stephan et al., 2019; Yang et al., 2020; Adebayo et al., 2021). While omics-based approaches, including spatial omics, have become increasingly prominent (Moses and Pachter, 2022; Williams et al., 2022; Fang et al., 2023), correlating gene expression data with precise cellular and subcellular

phenotypes remains challenging. Our quantitative phenotyping pipeline addresses this gap by sensitively detecting subtle morphological alterations, such as Golgi fragmentation during interphase without manual classification (Fig. 1 J). This approach refines previous classification methods (Nagao et al., 2020) and provides a detailed foundation for understanding cellular functions.

Golgi unlinking in late G2 crucially regulates the G2/M transition and mitotic progression in association with key cell cycle regulators such as AURKA and pericentriolar materials (Barretta et al., 2016; Rios, 2014). Consistent with previous studies (Cervigni et al., 2015; Mascanzoni et al., 2024), our phenotyping approach quantitatively captured subtle features associated with Golgi unlinking and its inhibition by the JNK2 inhibitor SP (Fig. S2). Notably, our analysis identified two subpopulations with distinct sensitivities to SP (Fig. 2 C). This cellular heterogeneity emphasizes the importance of single-cell analyses. SP treatment also disrupted the nucleolar localization of Ki-67 characteristic of the G2 phase (Solovjeva et al., 2012; van Schaik et al., 2022), suggesting that cell cycle arrest might precede Golgi unlinking defects (Fig. S2 B and cv_ki67 in Fig. S2 F). Although this observation does not directly contradict the Golgi unlinking checkpoint model, it suggests greater complexity in the G2/M transition. To gain a deeper understanding, comprehensive profiling of multiple cellular structures, as demonstrated in this study, will be essential.

Beyond cell cycle regulation, inter-organelle communication is a central focus (Bohnert and Schuldiner, 2018), as illustrated by the Golgi–centrosome axis in the Golgi checkpoint model (Barretta et al., 2016). To characterize Golgi morphology, we used the distance between the nucleus and Golgi as one of the indices (Fig. 1 C). The relative positioning of subcellular structures can provide insights into their functional relationships. Notably, we found a correlation between Golgi morphology and ciliation during the G0/G1 transition (Fig. 2, F and G). This quantitative approach can be extended to investigate the relative positions of centrosomes/cilia, Golgi, and nuclei during cell polarization and migration (Veland et al., 2014; Christensen et al., 2013; Rong et al., 2021; Frye et al., 2020) and organelle contacts involving the Golgi, ER, and mitochondria (Voeltz et al., 2024; Vallese et al., 2020; Kwak et al., 2020).

In the analysis of G0/G1 transition inhibition, we detected distinct changes in Golgi morphology (Fig. S3 E). AURKA, a key regulator of the cell cycle, is essential for ciliary disassembly and cell cycle reentry (Pugacheva et al., 2007; Plotnikova et al., 2012; Goto et al., 2013). Our findings align with these studies, further demonstrating that SP treatment has a similar effect (Fig. 3 B). In addition, inhibition of the G0/G1 transition resulted in a unique dispersion pattern of the Golgi, i.e., the center of mass remained near the nuclear periphery, contrasting with the dispersed Golgi observed in HeLa cells during G2 (Fig. S3 E, siAURKA and SP). This pattern resembles what has been reported in cycling hTERT-RPE1 cells depleted of AURKA (Kimura et al., 2018). Furthermore, a link between this interphase Golgi structure and centrosome integrity has been suggested (Kimura et al., 2018). Given that centrosomes and cilia share the centriole (basal body) as a core structure, it is plausible that Golgi and cilia cooperate during the G0/G1 transition. Although our analysis focusing specifically on the G0/G1 transition did not detect any significant disruption in the positional relationship between the Golgi and centrosomes (Fig. 4 B), further investigation of functional relationships such as material transport is likely to provide deeper insight.

The exact relationship between cell cycle arrest, ciliary disassembly, and Golgi dispersion remains unclear. These processes may be independent, as cell cycle reentry and ciliary disassembly do not always occur synchronously (Ford et al., 2018). Alternatively, AURKA, as a master regulator of these processes (Pugacheva et al., 2007; Plotnikova et al., 2012; Goto et al., 2013), may coordinate them. It remains to be determined whether Golgi dispersion in SP-treated or AURKA-depleted cells stems from disrupted cell cycle reentry, ciliary disassembly, or direct inhibition of AURKA (Fig. 4 C). If Golgi integrity is crucial for the G0/G1 transition, it could serve as a checkpoint (Fig. 4 C), similar to its role in the G2/M transition. However, unlike in the G2/M transition, where Golgi fragmentation promotes centrosomal accumulation and activation of AURKA (Persico et al., 2010), our results, consistent with a previous study (Kimura et al., 2018), suggest that AURKA contributes to the maintenance of Golgi integrity during the G0/G1 transition. The G0/G1 transition requires the accumulation of regulators such as AURKA (Pugacheva et al., 2007), Nek2 (Kim et al., 2015), and Nde1 (Kim et al., 2011) at the ciliary base, where they are locally regulated. Consistently, our analysis suggested that AURKA accumulation at the centrosome is essential for the G0/G1 transition (Fig. 4 A). Additionally, Golgi trafficking is closely associated with ciliary formation and maintenance (Masson and El Ghouzzi, 2022; Stevenson, 2023; Jin et al., 2022). Therefore, an intact Golgi may be essential for transporting these molecules. The appearance of SP-specific clusters even before serum re-addition (Fig. 2 H) suggests that irregular Golgi dispersion may influence these processes, opening an exciting avenue for future research.

In conclusion, we present an image-based cell phenotype profiling method to analyze cellular and subcellular morphological features for insights into cell functions. This method enabled us to detect subtle changes related to the G2/M transition, mitosis, and the G0/G1 transition, providing datasets that directly enable functional analysis. Specifically, we identified a unique morphological change in the Golgi during G0/G1 transition inhibition, coinciding with persistent cilia retention. Our findings suggest that the Golgi-cilia axis may play an integral role in the G0/G1 transition, warranting deeper investigation. By integrating this study with cell biology assays and omics approaches, we can advance our understanding of cell cycle regulation and move toward constructing a comprehensive cell phenotype database.

# Materials and methods

## Antibodies

The following primary antibodies were used for IF: α-Tubulin (1: 500; mouse monoclonal, clone 1E4C11, 66031; Proteintech), GM130 (1:500; mouse monoclonal, clone 35, 610822; BD Biosciences), Ki-67 (1:500; rabbit polyclonal, 27309; Proteintech), acetylated α-tubulin (1:10,000; mouse monoclonal, clone 6-11B-1, MABT868; Merck), GRASP65 (1:500; rabbit monoclonal, clone EPR12439, ab174834; Abcam), centrin (mouse monoclonal, clone 20H5, 1:500; 04-1624; Merck), γ-tubulin (1:1,000; rabbit polyclonal, T5192; Merck), and AURKA (1:200; rabbit monoclonal, 14475; CST). For western blotting (WB), antibodies included AURKA (1:500; rabbit polyclonal, 28371-1-AP; Proteintech), JNK1 (1:1,000; rabbit polyclonal, 51151-1-AP; Proteintech), JNK2 (1: 1,000; rabbit polyclonal, 51153-1-AP; Proteintech), and β-actin (1: 2,000; rabbit monoclonal, AF5003; Beyotime). Secondary antibodies conjugated with Alexa Fluor 568 or 647 (goat polyclonal, A-11004, A-11011, A-21235, A-21244; Thermo Fisher Scientific) were used at 1:500 for IF, and an HRP-conjugated (horseradish peroxidase) secondary antibody (goat polyclonal, A0208; Beyotime) was used at 1:2,000 for WB.

## Chemicals and reagents

The following reagents were used in cell culture or treatment: DAPI (400 ng/ml; D8417; Sigma-Aldrich) and Alexa Fluor 488–conjugated phalloidin (1:800; A12379; Thermo Fisher Scientific) for fluorescence staining; thymidine (2 mM; T1895; Sigma-Aldrich) and ProTAME (12 μM; 1362911; R&D Systems) for cell cycle synchronization; taxol (50 nM for mitosis or 1 μM for G2; HY-B0015; MedChemExpress), monastrol (100 nM; HY-101071A; MedChemExpress), SP600125 (50 μM; S5567; Sigma-Aldrich),

cytochalasin D (100 ng/ml; C8273; Sigma-Aldrich), and brefeldin A (200 ng/ml; B7651; Sigma-Aldrich) for pharmacological inhibition. EdU (Alexa Fluor 594–conjugated, C10339; Thermo Fisher Scientific) was used for labeling proliferating cells. Collagen I (7 µg/cm$^2$; 354236; Sigma-Aldrich) was used for coating culture surfaces.

### siRNAs

Cells were transfected with siRNAs targeting JNK1 (5′-GTGGAA AGAATTGATATATAA-3′; Tsingke Biotechnology), JNK2 (5′-AAG AGAGCTTATCGTGAACTT-3′; Tsingke Biotechnology), or AURKA (5′-AUGCCCUGUCUUACUGUCA-3′; Tsingke Biotechnology).

### Cell lines

HeLa cells were obtained from the China Center for Type Culture Collection (GDC0009), and ARPE-19 cells were obtained from the National Collection of Authenticated Cell Cultures, China (GNHu45).

### Cell culturing and IF

Cell culturing and IF methods were previously described (Takao et al., 2017; Takao et al., 2019). Briefly, HeLa cells were cultured in DMEM supplemented with 10% FBS (AUS-01S; Cell-Box) and 1% penicillin/streptomycin. ARPE-19 cells were cultured in DMEM/F12 supplemented with 10% FBS (AUS-01S; Cell-Box) and 1% penicillin/streptomycin. For serum-starvation of ARPE-19 cells, cells were washed with serum-free DMEM/F12 twice to remove residual serum, then cultured in serum-free DMEM/F12. For cell cycle synchronization in S phase, cells were seeded on coverslips coated with collagen I, and 2 mM thymidine was added to the medium, as indicated in Fig. S1 D and Fig. S2 A. Many cells entered mitosis around 10 h after the second release from thymidine block, which determined the time points for mitosis and late G2 analyses (Fig. S1 D and Fig. S2 A). ProTAME (12 µM) was then added to further arrest cells at metaphase (Fig. S1 D).

Cells were transfected with siRNA using Lipofectamine RNAiMAX according to the manufacturer's instructions. We confirmed sufficient reduction of target protein levels before proceeding with subsequent experiments (see Fig. S5 B and the WB section for details).

Cells grown on coverslips were fixed with 4% paraformaldehyde (PFA) in PBS for 10 min at RT or with methanol for 5 min at –20°C (only when using antibodies against centrin or γ-tubulin). Cells were then incubated in blocking buffer (0.05% Triton X-100 and 1% BSA in PBS) for 20 min at RT to permeabilize and block. The cells were then incubated with primary antibodies for 1 h at RT, washed three times with PBS, and then incubated with secondary antibodies for 1 h at RT. Phalloidin–Alexa Fluor 488 was added together with the secondary antibodies to label actin filaments. After three additional washes, cells were incubated with DAPI in PBS for 5 min at RT, washed again, and mounted using ProLong Gold (P36930; Thermo Fisher Scientific).

### WB

Cells were lysed in RIPA buffer (Beyotime) containing a protease inhibitor cocktail (AbMole) on ice for 30 min, followed by centrifugation at 21,500 × *g* for 10 min at 4°C. The supernatant was mixed with 5 × SDS loading buffer (Yeasen) and boiled at 100°C for 10 min. Proteins were separated by SDS-PAGE using Glass-PAGE HEPES–Tris gels (WanSheng HaoTian) and transferred onto PVDF membranes (Millipore). Membranes were blocked with 5% skim milk (Biosharp) in TBST for 2 h at RT, incubated with primary antibodies overnight at 4°C, and then with HRP-conjugated secondary antibodies (Beyotime) for 1 h at RT. Signals were detected using a chemiluminescent HRP substrate (Millipore) and imaged with a Tanon 5200 system.

For quantification of protein levels, western blot images were analyzed using ImageJ (Schneider et al., 2012). Background subtraction was performed using the rolling ball algorithm with a radius of 50.0 pixels, and individual bands were selected using the wand tool. The integrated density of each band was measured, and protein levels were normalized to actin.

### Microscopy

An inverted confocal microscope (Nikon, AXR NSPARC) equipped with a 40× water-immersion objective (Nikon, CFI Apochromat LWD Lambda S 40XC WI, NA 1.15) was used for image acquisition at RT. Using the microscope's operating software NIS-Elements, z-stack confocal images were acquired at 1.0- or 0.5-µm intervals, with the pixel sizes of 0.4316 µm (1.0× zoom) and 0.2877 µm (1.5× zoom), for HeLa and ARPE-19 cells, respectively.

### Image processing and analysis

All confocal images, containing multiple channels and z-slices, were saved in Nikon NIS-Elements ND2 format (12-bit depth) and converted to 16-bit multichannel TIFF files with maximum intensity projection using Fiji (Schindelin et al., 2012). These TIFF files served as the "original images" for subsequent processing and analysis in Python on the Visual Studio Code platform. All the code was written in Jupyter (IPython) notebook format and is available in Dryad (see Data availability). For details of the processing and analysis, also see the main text and these codes with notes. Some key steps are described below.

The original images were converted to RGB for cellpose segmentation. By overly enhancing the contrast, α-tubulin or actin filament images were used as cytoplasmic markers for segmentation. For the mammalian cell lines we used in this study (HeLa and ARPE-19, as well as PK-15 in unpublished work), cellpose (version 2) with the cyto2 model successfully segmented cells across varying densities, from sparse to confluent cultures. Each segmented cell was assigned an identifier (e.g., cells #38 and #81 in Fig. S1 A). Note that RGB conversion and contrast enhancement were performed solely for segmentation, whereas the original 16-bit images were used for quantitative analyses to avoid information loss.

Subcellular structures such as nuclei, Golgi, and cilia were segmented by binarization using Otsu's method (OpenCV) or Yen's method (scikit-image). The "locate" function of the Trackpy package was used to extract fluorescence peaks, and the "skeletonize" function of the scikit-image package was used to skeletonize objects. Cell phenotype profiles in Pandas DataFrame format were exported as CSV files after measuring various features.

Cell phenotype profiles generated from multiple images were then combined into a single DataFrame and standardized by robust z-score scaling (RobustScaler, scikit-learn). UMAP and DBSCAN analyses were performed using the umap-learn and scikit-learn packages, respectively. Plots were created mainly with matplotlib and seaborn; in some cases, the "RainCloud" function from the PtitPrince package was used.

### Data availability

The datasets (datasets 1–7), original image data, cell profile data, and analysis code have been deposited in Dryad and are available at https://doi.org/10.5061/dryad.8gtht771s.

## Acknowledgments

We gratefully acknowledge Yasushi Okada for invaluable advice; Takumi Chinen for continuous, fruitful discussions; Shuhong Zhao and members of their research group for support and insightful discussions; Chunli Chen for providing Yiming Peng with the opportunity to participate in this project; Honghong Zhou and Zhixiang Xu for contributions to establishing the initial research environment; and members of the Takao lab for technical support and discussions. Confocal microscopy was performed at the Public Instrument Center of the College of Animal Science and Technology and College of Veterinary Medicine at Huazhong Agricultural University.

This work was supported by grants from the National Natural Science Foundation of China (32350610255 and 32341051).

Author contributions: Xun Cao: data curation, formal analysis, investigation, resources, software, validation, and writing—review and editing. Yiming Peng: conceptualization, data curation, formal analysis, investigation, resources, software, validation, and writing—review and editing. Mengyuan Yang: data curation, formal analysis, investigation, resources, validation, and writing—review and editing. Mengling Gan: data curation, resources, validation, and writing—review and editing. Di Zhang: investigation, resources, validation, and writing—review and editing. Shiyue Zhou: conceptualization and data curation. Daisuke Takao: conceptualization, data curation, formal analysis, funding acquisition, investigation, methodology, project administration, resources, software, supervision, validation, visualization, and writing—original draft, review, and editing.

Disclosures: The authors declare no competing interests exist.

Submitted: 25 March 2025

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

**Supplemental material**

Figure S1. **Morphological features of cellular and subcellular structures for cell phenotype profiling. (A)** Example of the cell segmentation process and extraction of single cells. HeLa cells were segmented using contrast-enhanced images of microtubules and DAPI-stained nuclei. Each cell was assigned an identifier, and two representative cells (#38 and #81) are shown. Scale bar, 50 µm. **(B and C)** Examples of abstraction-based analysis of Golgi structure. Golgi morphology was characterized from GM130 images by analyzing fluorescence peaks (B) and by extracting skeletonized line objects (C). The boxed regions in the nucleus–Golgi distance measurements correspond to the cropped areas shown in Fig. 1 C. Scale bar, 50 µm. **(D)** Timeline for the analysis of mitotic spindle defects in metaphase-arrested HeLa cells. HeLa cells were synchronized with a double thymidine block, followed by ProTAME treatment to accumulate cells in metaphase. To induce specific spindle assembly defects, cells were treated with DMSO (control), taxol, or monastrol for 6 h, when most cells were expected to be in G2 phase. **(E)** Representative cell images. The left panel displays a full image of DMSO-treated cells, while the right panel shows magnified views of typical mitotic cells. Scale bars: 50 µm (full image) and 20 µm (magnified image). **(F)** Correlation matrix of all features used for cell phenotype profiling, as shown in Fig. 1 E. See Table S1 for descriptions of individual features.

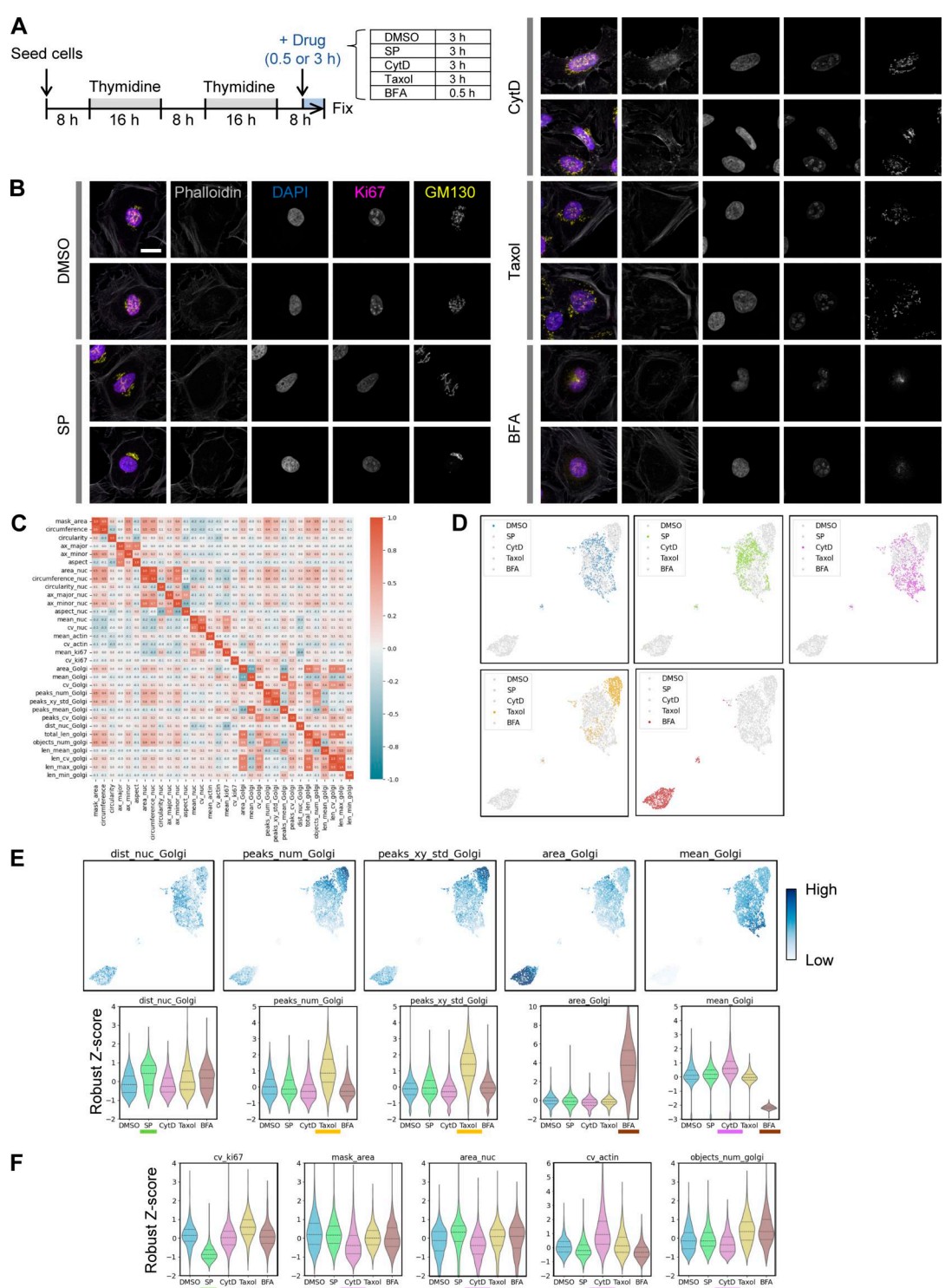

Figure S2. **Characterization of drug treatment effects on cellular and subcellular morphology in late G2. (A)** Timeline for cell cycle synchronization using a double thymidine block followed by drug treatment. HeLa cells were used; at the time of fixation, most cells had reached late G2, just before entering mitosis. **(B)** Representative images of HeLa cells in late G2 fixed and stained after drug treatment. Cropped images of representative cells from each experimental condition are shown. Scale bar, 20 μm. **(C)** Correlation matrix for all features used in the cell phenotype profiling shown in Fig. 2 B. See Table S1 for detailed descriptions of each feature. **(D)** UMAP plots of cell phenotype profiles. The UMAP data from Fig. 2 B are shown, with each condition highlighted individually for clarity. **(E)** Selected features that robustly reflect the effect of drug treatment on Golgi morphology in late G2. Drug names associated with a strong effect on each feature are underlined in color. Features include dist_nuc_Golgi (distance between the centers of mass of the nucleus and Golgi) for SP, peaks_num_Golgi and peaks_xy_std_Golgi (number and spatial variance of GM130/Golgi fluorescence peaks) for taxol, and area_Golgi and mean_Golgi (area and mean fluorescence intensity of the GM130/Golgi region) for BFA and CytD. Full data are provided in dataset 3. **(F)** Notable features in addition to those related to Golgi morphology. These include cv_ki67 (CV for nuclear Ki-67 fluorescence intensity, i.e., SD normalized by the mean) for SP and mask_area (cell mask area) and cv_actin (CV of phalloidin/actin staining) for CytD. Full data are provided in dataset 3.

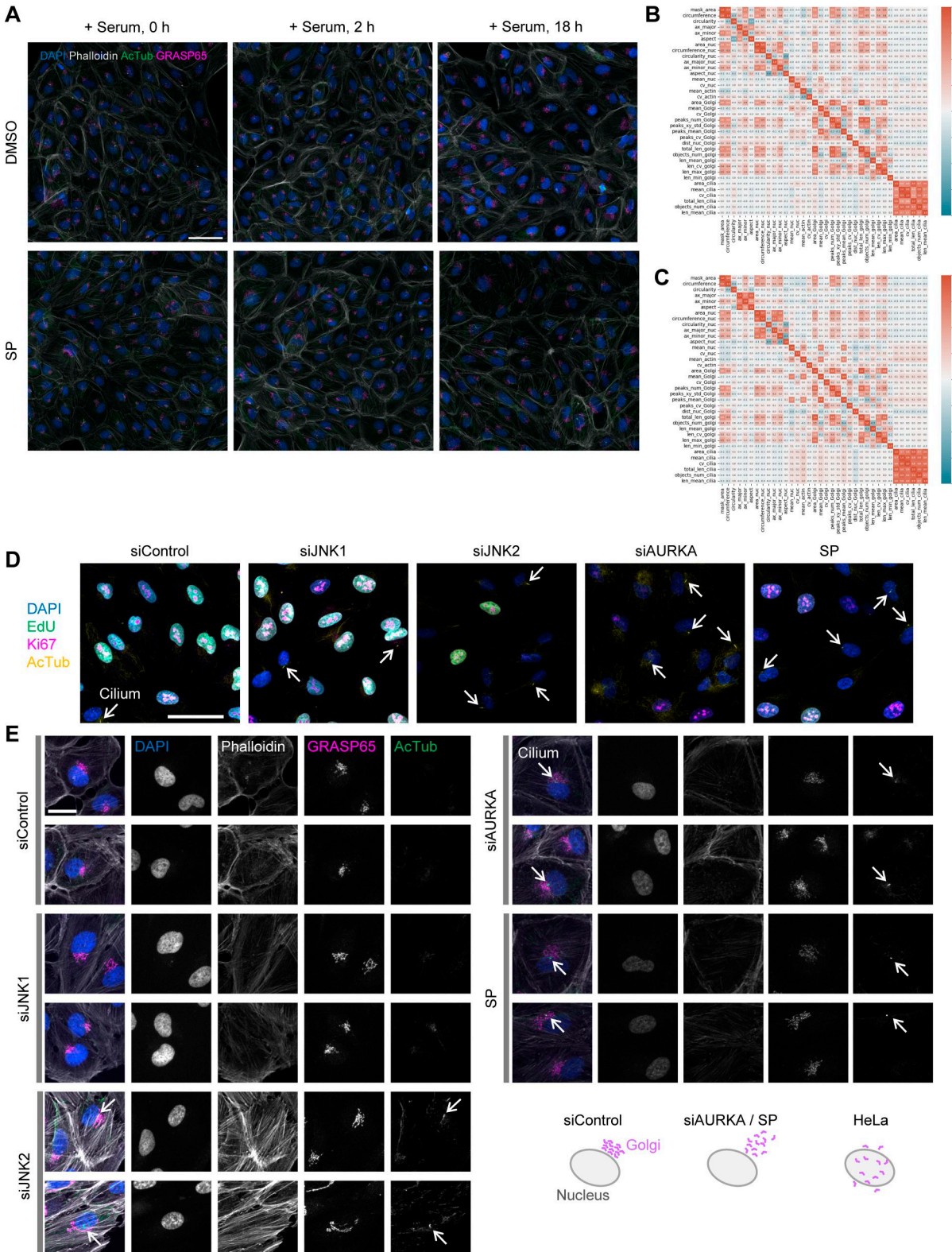

Figure S3. **Representative cell images after serum re-addition and features used in analysis. (A)** Representative images of ARPE-19 cells for the indicated experimental conditions and time points. Scale bar, 50 µm. **(B and C)** Correlation matrices of the features used in cell phenotype profiling in Fig. 2 H and Fig. 3 D, respectively. See Table S1 for detailed feature descriptions. **(D)** Representative fluorescence images used to monitor cell cycle progression and ciliation. Images of ARPE-19 cells were acquired 18 h after serum re-addition (see Fig. 2 E), stained for EdU (via click reaction), Ki-67 (IF), and acetylated tubulin (AcTub; IF). The analysis results are shown in Fig. 3 B. Scale bar, 50 µm. **(E)** Representative cropped images of ARPE-19 cells used for cell phenotype profiling, related to those shown in Fig. 3 C. Arrows indicate cilia. AcTub, acetylated tubulin. Scale bar, 20 µm. Schematics illustrate typical Golgi morphology in the siControl group, siAURKA- or SP-treated group, and HeLa cells in late G2 (see Fig. S2 B, DMSO).

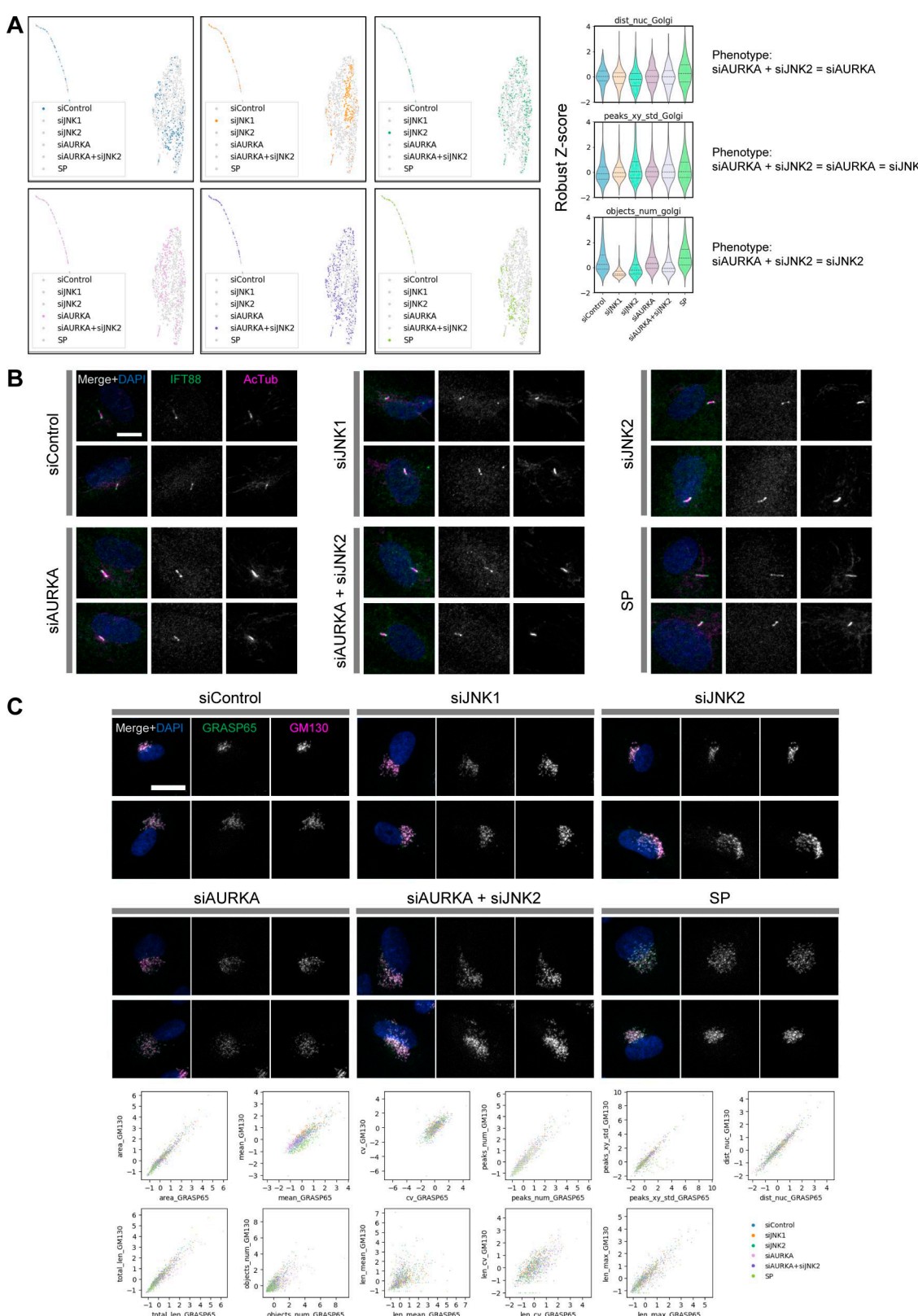

Figure S4. **Analysis of double gene knockdown and marker validation at the G0/G1 transition. (A)** Same analysis as in Fig. 3, D and E, but with the addition of double knockdown of AURKA and JNK2. Full data are available in dataset 7. **(B and C)** Co-staining analysis of cilia and Golgi. Cilia and Golgi were simultaneously stained with antibodies against two respective marker proteins, following the time course shown in Fig. 3 A (including double knockdown of AURKA and JNK2). Representative cropped images of (B) cilia markers IFT88 and acetylated tubulin (AcTub) and (C) Golgi markers GRASP65 and GM130 are shown. Scale bars: (B) 10 µm and (C) 20 µm. For the Golgi markers, the same morphological features were extracted from images of each marker and compared (see plots at the bottom).

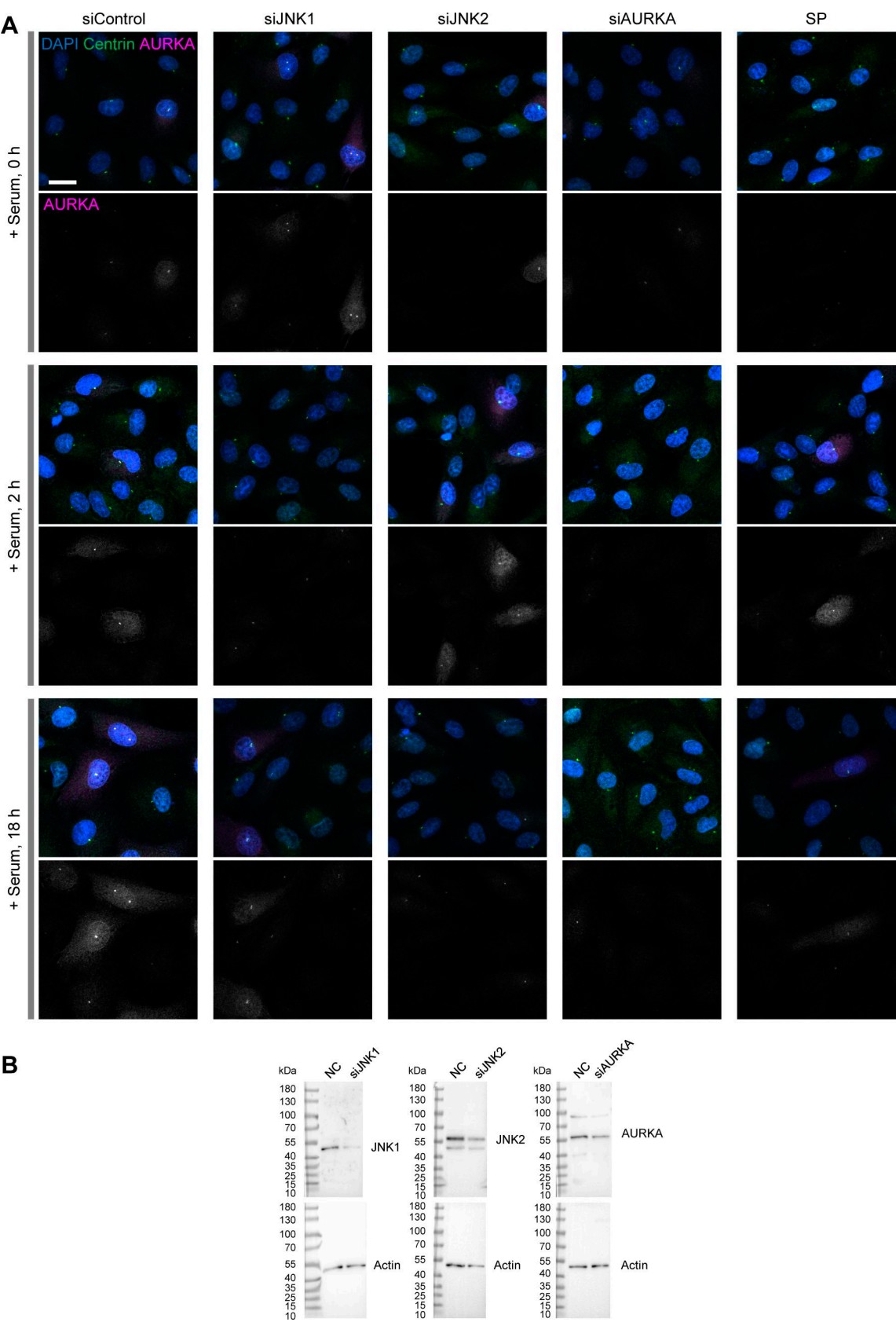

Figure S5. **Analysis of AURKA accumulation at centrosomes and evaluation of gene knockdown efficiency. (A)** Representative images used for the analysis of AURKA accumulation at centrosomes, related to Fig. 4 A. Scale bar, 20 μm. **(B)** Western blot analysis to evaluate gene knockdown efficiency. Relative to the negative control (set as 100%), protein expression levels were reduced to 16.6% (siJNK1), 41.5% (siJNK2), and 38.6% (siAURKA), respectively. Source data are available for this figure: SourceData FS5.

**Provided online is Table S1. Table S1 shows list of features used for cell phenotype profiling.**

