## [Peer Review File · The Journal of Cell Biology]

High-Content Phenotyping Reveals Golgi Dynamics and Their Role in Cell Cycle Regulation

Xun Cao, Yiming Peng, Mengyuan Yang, Mengling Gan, Di Zhang, Shiyue Zhou, and Daisuke Takao

Corresponding Author(s): Daisuke Takao, Huazhong Agricultural University

Review Timeline:

Submission Date:	2025-03-25
Editorial Decision:	2025-06-02
Revision Received:	2025-08-01
Editorial Decision:	2025-09-15
Revision Received:	2025-09-28
Editorial Decision:	2025-10-03
Revision Received:	2025-10-08

Monitoring Editor: Alexey Khodjakov

Scientific Editor: Dan Simon

Transaction Report:

DOI: <https://doi.org/10.1083/jcb.202503083>

June 2, 2025

Re: JCB manuscript #202503083

Daisuke Takao
Huazhong Agricultural University

Dear Dr. Takao,

Thank you for submitting your manuscript entitled "High-Content Phenotyping Reveals Golgi Dynamics and Their Role in Cell Cycle Regulation." The manuscript has been evaluated by expert reviewers, whose reports are appended below. Unfortunately, after an assessment of the reviewer feedback, our editorial decision is against publication in JCB of the current version of your manuscript.

The Reviewers recognize the power of the described workflow and methodology for unbiased morphological profiling of large cell populations. There is a significant demand for high-throughput morphological analyses and thus the 'Tool' is likely to be appreciated by many cell biologists. However, the Reviewers also identify two major deficiencies in the current manuscript: 1. The study provides minimal insight into molecular mechanisms related to the Golgi and/or cilia; and 2. The capabilities and robustness of the described approach are not sufficiently validated. While relatively limited novelty of mechanistic findings is not a principal obstacle for a 'Tool' submission, a deeper characterization of capabilities, robustness, and limitation of the described approach are essential for us to consider a revised manuscript. The Reviewers, particularly Reviewer 2 offer several excellent suggestions on how the work can be strengthened.

Given interest in the topic, we would be open to resubmission to JCB of a significantly revised and extended manuscript that fully addresses the reviewers' concerns and is subject to further peer-review. While we recognize that not all of the suggested additions will be incorporated into the revised manuscript, a detailed response to all Reviewers' concerns and the reasons for choosing specific experimental additions to your work must accompany the revision.

If you would like to resubmit this work to JCB, please contact the journal office to discuss an appeal of this decision or you may submit an appeal directly through our manuscript submission system. Please note that priority and novelty would be reassessed at resubmission.

Regardless of how you choose to proceed, we hope that the comments below will prove constructive as your work progresses. We would be happy to discuss the reviewer comments further once you've had a chance to consider the points raised in this letter. You can contact the journal office with any questions at cellbio@rockefeller.edu.

Thank you for thinking of JCB as an appropriate place to publish your work.

Sincerely,

Alexey Khodjakov, PhD
Monitoring Editor
Journal of Cell Biology

Dan Simon, PhD
Scientific Editor
Journal of Cell Biology

Reviewer #1 (Comments to the Authors (Required)):

This is a study concentrating on the quantification of cellular features in fixed-cell populations in the relation to the cell cycle. This method allowing for correlation of cell morphology and organelle features in large data sets would be very useful of the field.

However, biologically, the paper is marginally novel: biological findings reported here as new have been observed and reported. The significance of observations is overstated in the abstract and discussion.

For example, Golgi fragmentation in taxol was first reported at least in 1983 (<https://pubmed.ncbi.nlm.nih.gov/6136036/>) and many times since. Further, the change of Golgi morphology and Golgi-centrosome distance in G1/S/G2 transition has been described in detail (<https://pubmed.ncbi.nlm.nih.gov/32344866/>). Increase of the nuclear size upon G2 arrest also has been

reported and its interpretation is obvious. The observation that Aurora A inhibition promotes Golgi dispersion is also not novel (e.g. <https://pubmed.ncbi.nlm.nih.gov/29571950/> etc). It is possible that the authors meant to emphasize that their method is a great tool to study those previously reported phenomena, but this is not how the article reads. This has to be corrected.

Some conclusions are overstretched: for example, the intensity of a single Golgi-associated protein cannot be interpreted as a measure of "Golgi fragmentation": only a single protein has been detected and its concentration at the Golgi membrane may vary in different conditions. It has to be proposed to use more than one Golgi marker, and this complication has to be discussed.

Importantly, we recommend the authors to read the papers that they cite in detail for proper usage of references when they talk about quantifying phenomena previously observed by other groups (e.g. Frye et al 2023 measures not the Golgi volume, but rather Golgi morphology details exceeding what is described here, albeit not for large data sets).

What this paper shows, is that this method is, indeed, a quite valuable tool to detect the changes described previously. It has to be published as a methods/technology paper and not as a JCB research article.

Reviewer #2 (Comments to the Authors (Required)):

In this manuscript, Cao et al. present a workflow for quantitatively analyzing immunofluorescence images to generate single-cell phenotypic profiles based on various cellular and suborganelle features. To validate the method, they focus primarily on the morphology of the Golgi apparatus across different cell cycle stages and under treatment with Taxol, BFA, CytD, and a small molecule SP inhibitor. Using this pipeline, they can distinguish and cluster cells with compact, dispersed, and fragmented Golgi phenotypes.

To further demonstrate the method's resolution, the authors use a starvation-induced ciliogenesis assay to assess the impact of SP inhibition on the G0-G2 transition. While most control cells reabsorb their cilia by 18 hours after serum re-addition, around 20% of SP-treated cells remain ciliated.

The proposed pipeline is potentially useful for unbiased single-cell analysis, and the multivariate approach combined with clustering allows for efficient identification of distinct cellular subpopulations. However, while the methodology may be of interest, the manuscript lacks critical controls and does not offer significant biological insights or novel mechanistic findings. Further validation and a deeper interpretation of the observations are required for publication in JCB.

Major comments:

- The main strength of the pipeline is its ability to extract phenotypic features at the single-cell level and associate multiple parameters (e.g., Golgi morphology and nuclear area) under specific conditions using DBSCAN-based clustering. However, the biological findings themselves, such as Taxol-induced Golgi fragmentation, are already well established. The phenotype observed in SP-treated cells during the G0-G2 transition deserves deeper investigation. For instance, AURKA has been shown to be essential for cilia disassembly in RPE cells (Pugacheva et al., Cell, 2007) and is also implicated in Golgi fragmentation (Persico et al., MBoC, 2010). Given this, could your pipeline be used to analyze AURKA localization relative to the centrosome or Golgi? Could you further link AURKA inhibition with Golgi fragmentation during G0-G1 transition? The mechanism presented here lacks clarity and detail.

- I have concerns regarding the ciliogenesis assays. The authors acknowledge that their observed ciliation rates are lower than previously reported (40% in their work vs. 70% in Wang & Brautigan, 2008, or 80% in Partisani et al., 2021). Is the cell line employed the best for this type of study?

Also, is confluency consistent across all experiments? For example, in Figure S7A, the top panels for serum +2 h and +18 h appear less confluent. Could variation in confluency affect ciliogenesis outcomes?

- Would using an additional cilia marker, such as Arl13B, yield similar results? Acetylation may be affected, but not the cilium itself.

- siRNA experiments would benefit from data showing knockdown efficiency. A western blot or quantification panel should be included.

Minor comments:

- One focus of the study is Golgi unlinking during G2, yet the only synchronization method used is a double thymidine block. How do you confirm that cells are in G2, and particularly late G2? The inclusion of cell cycle markers would strengthen this claim. Since your method allows single-cell analysis, why not include a G2 marker (e.g., CENP-F) in an unsynchronized population to assess Golgi morphology more accurately?

- There appear to be duplicated images: Figures 2B and S4B, as well as Figures 4C and S9B, show identical representative fields. Please revise accordingly.

- In the ciliogenesis and G0-G2 transition figures, cilia are difficult to visualize. Separating the acetylated tubulin channel may improve clarity.

- Depletion of JNK1 does not appear to affect the G0-G2 transition, while JNK2 and AURKA knockdowns phenocopy the SP inhibitor. Would you expect additive or synergistic effects from the co-depletion of JNK2 and AURKA?

Reviewer #3 (Comments to the Authors (Required)):

The authors present a study that attempts to use image-based morphological profiling to study the functions of the Golgi and cilia during the cell cycle. However, the work is largely descriptive, relying on morphometric data from low-resolution optical images of cells subjected to treatments such as small molecule and serum withdrawal. The manuscript suffers from several significant issues:

- 1) The conclusions are unconvincing, as the reported morphological phenotypes are, at best, only tangentially related to the underlying molecular and cellular mechanisms governing Golgi and cilia.
- 2) While some morphological changes are detected through quantitative profiling, their functional significance remains unclear and is not further experimentally explored.
- 3) Many hypotheses and claims lack experimental support. For instance, there is no direct evidence provided to substantiate the claim of Golgi unlinking.

Overall, the study does not provide novel mechanistic insights into Golgi and cilium biology and is unlikely to make a significant impact on the broader field of cell biology. Its relevance may be limited to a niche audience focused on high-throughput morphological screening.

Response to reviewer comments

We sincerely thank all reviewers for taking the time and effort to evaluate our manuscript. We believe that the addition of new experimental data and substantial revisions to the text have greatly improved the quality of our work. We begin here with a general response.

Based on the reviewers' comments, we understand that the major concerns consistently fall into the following three key points:

1. Technical utility and dataset contribution

All reviewers recognized the utility of our methodology and the value of the dataset we provided, and no technical flaws were raised. Given that our initial submission was under the Tools category, we regard this as highly positive feedback. In recent years, especially with the rapid rise of bioinformatics, top-tier laboratories have increasingly dominated high-impact studies through sheer scale. In contrast, our small group, comprising only a PI and students, established this method and generated a large dataset. Even students with little prior experience in image analysis or programming were able to master the analytical techniques used in this study within a few months and apply them to their own projects. This low barrier to entry and high performance are important features that we believe can contribute to the advancement of more inclusive and idea-driven research, as well as to a more open scientific culture. We therefore believe our work is well aligned with the mission of JCB's Tools category.

2. Limited biological novelty

We fully acknowledge that, even for a Tools paper, the journal requires a certain level of biological insight. In response, we conducted additional experiments to further investigate the role of the Golgi-cilia axis in cell cycle regulation through AURKA. We have incorporated almost all of the additional experiments suggested by the reviewers into the revised manuscript. While we plan to pursue a more in-depth analysis of the G0/G1 transition as a separate project, we believe the revised manuscript demonstrates sufficient biological significance within the scope of a methods-focused study.

3. Previously known phenotypes

We acknowledge that some of the phenotypes detected by our analysis, particularly those in the G2 phase, are not novel. In fact, we initially validated our analytical framework by targeting phenotypes with dramatic changes, such as defective mitotic spindle formation, followed by more subtle and challenging targets like Golgi unlinking in G2. Our intention was never to claim discovery of these known features. In response to the reviewers' valid concerns, we have extensively revised the figures and text to ensure that these validation results are clearly presented as analyses of known phenotypes.

We have extensively revised the manuscript, including the addition of new data and the reorganization of several figures. To improve clarity and readability, we have also prepared to make the full dataset publicly available through an external repository, rather than including it as supplementary figures. Major revisions in the manuscript are highlighted with markers. In addition, paragraphs newly added to accommodate the new data are indicated in blue for ease of reading. We believe the revised manuscript fully addresses the reviewers' concerns, but we welcome further discussion if needed. Below, we provide detailed point-by-point responses to each comment.

Reviewer #1:

Comment:

This is a study concentrating on the quantification of cellular features in fixed-cell populations in the relation to the cell cycle. This method allowing for correlation of cell morphology and organelle features in large data sets would be very useful of the field.

However, biologically, the paper is marginally novel: biological findings reported here as new have been observed and reported. The significance of observations is overstated in the abstract and discussion.

Response:

We sincerely acknowledge the concern regarding overstatement and have thoroughly revised the abstract and discussion to more accurately reflect the biological significance of our findings.

Comment:

For example, Golgi fragmentation in taxol was first reported at least in 1983 (<https://pubmed.ncbi.nlm.nih.gov/6136036/>) and many times since. Further, the change of Golgi morphology and Golgi-centrosome distance in G1/S/G2 transition has been described in detail (<https://pubmed.ncbi.nlm.nih.gov/32344866/>). Increase of the nuclear size upon G2 arrest also has been reported and its interpretation is obvious. The observation that Aurora A inhibition promotes Golgi dispersion is also not novel (e.g. <https://pubmed.ncbi.nlm.nih.gov/29571950/> etc). It is possible that the authors meant to emphasize that their method is a great tool to study those previously reported phenomena, but this is not how the article reads. This has to be corrected.

Response:

In response to this and similar comments from the other reviewers, we have thoroughly revised all relevant sections of the manuscript to clarify our intentions and avoid misinterpretation (**highlighted with markers**). Specifically, the analyses of mitotic cells and late G2 were conducted primarily to evaluate the performance of our method using well-established phenotypes, such as taxol-induced Golgi fragmentation (e.g., lines 12–16 on page 9, lines 16–17 on page 10, and lines 1–5 on page 12 of the revised manuscript). We have now taken care to explicitly state this purpose to avoid any misunderstanding that these results present novel biological discoveries.

As previously acknowledged in the original version, we are aware of prior studies describing Golgi morphological changes and Golgi-centrosome distance across the G1/S/G2 phases. However, compared to the extensively studied dynamics during late G2 to mitotic exit, the morphological behavior of the Golgi apparatus during interphase, particularly in the context of ciliary disassembly and cell cycle re-entry, has received much less attention. Our work focuses specifically on this underexplored phase.

We also recognize that Golgi dispersion upon Aurora A inhibition has been reported, including the study by Kimura et al. (2018), which we had already cited and discussed in the previous version (corresponds to lines 17–21 on page 21 of the revised manuscript). While Kimura et al. examined Golgi changes over broader phase comparisons (i.e., interphase vs. mitosis), our study centers on the G0/G1 transition. We believe this context matters. Additionally, our manuscript (Fig S3E) quantitatively illustrates that Golgi dispersion upon AURKA inhibition is distinct from common dispersion patterns, highlighting variability that may have biological significance, although further investigation is required.

We hope these revisions clarify the scope and intention of our study. If additional clarification is needed, we would be happy to engage in further discussion.

Comment:

Some conclusions are overstretched: for example, the intensity of a single Golgi-associated protein cannot be interpreted as a measure of "Golgi fragmentation": only a single protein has been detected and its concentration at the Golgi membrane may vary in different conditions. It has to be proposed to use more than one Golgi marker, and this complication has to be discussed.

Response:

We added co-staining results with two Golgi markers, GM130 and GRASP65, and demonstrated the co-localization of these markers and high correlation of the independently acquired features from each channel (Fig S4C and lines 3–9 on page 18 of the revised manuscript). While it is common in many studies to discuss Golgi morphology using only a single marker, we appreciate the reviewer's point that validation using multiple markers is particularly important in our manuscript, which focuses on quantitative analysis.

Additionally, we acknowledge the concern about relying solely on fluorescence intensity measurements. Given this limitation, we have adopted a comprehensive analytical approach that integrates multiple parameters in this study. However, we also agree with the reviewer that, even though our approach integrates multiple morphological features, these features are still ultimately derived from detected fluorescence signals. Therefore, we fully support the recommendation to validate findings using multiple Golgi markers, and we have addressed this point in the revised manuscript as described above (Fig S4C and lines 3–9 on page 18 of the revised manuscript). Thank you for highlighting this important aspect.

Comment:

Importantly, we recommend the authors to read the papers that they cite in detail for proper usage of references when they talk about quantifying phenomena previously observed by other groups (e.g. Frye et al 2023 measures not the Golgi volume, but rather Golgi morphology details exceeding what is described here, albeit not for large data sets).

Response:

As a point of clarification, Frye et al. (2023) do report measurements of Golgi volume. We highly value their focus on the three-dimensional structure of the Golgi, which is why we emphasized that aspect in the original version of our manuscript. Nonetheless, we fully acknowledge that their study also quantified additional features such as Golgi morphology and its spatial relationship with the centrosome. We have now revised the relevant section (lines 5–10 on page 6 of the revised manuscript) to reflect this more accurately.

We would also like to stress that our citations of Frye et al. and other studies involving quantitative analysis of the Golgi were made in a positive context, highlighting them as strong examples in contrast to the many studies relying solely on qualitative descriptions. If any of our wording was interpreted as suggesting a critique of these important prior works, that was certainly not our intention. We have revised the text accordingly to prevent such misunderstanding (lines 5–10 on page 6 of the revised manuscript).

Comment:

What this paper shows, is that this method is, indeed, a quite valuable tool to detect the changes described previously. It has to be published as a methods/technology paper and not as a JCB research article.

Response:

We greatly appreciate the acknowledgment that our method is a valuable tool for detecting changes in intracellular structures. With the additional analyses included in the revised manuscript, we believe our study is now well-suited for publication as a JCB Tools article.

Reviewer #2:

Comment:

In this manuscript, Cao et al. present a workflow for quantitatively analyzing immunofluorescence images to generate single-cell phenotypic profiles based on various cellular and suborganelle features. To validate the method, they focus primarily on the morphology of the Golgi apparatus across different cell cycle stages and under treatment with Taxol, BFA, CytD, and a small molecule SP inhibitor. Using

this pipeline, they can distinguish and cluster cells with compact, dispersed, and fragmented Golgi phenotypes.

To further demonstrate the method's resolution, the authors use a starvation-induced ciliogenesis assay to assess the impact of SP inhibition on the G0-G2 transition. While most control cells reabsorb their cilia by 18 hours after serum re-addition, around 20% of SP-treated cells remain ciliated.

The proposed pipeline is potentially useful for unbiased single-cell analysis, and the multivariate approach combined with clustering allows for efficient identification of distinct cellular subpopulations. However, while the methodology may be of interest, the manuscript lacks critical controls and does not offer significant biological insights or novel mechanistic findings. Further validation and a deeper interpretation of the observations are required for publication in JCB.

Response:

Thank you for your specific and constructive comments. Below, we provide point-by-point responses to address the concerns raised.

Comment:

Major comments:

-The main strength of the pipeline is its ability to extract phenotypic features at the single-cell level and associate multiple parameters (e.g., Golgi morphology and nuclear area) under specific conditions using DBSCAN-based clustering. However, the biological findings themselves, such as Taxol-induced Golgi fragmentation, are already well established. The phenotype observed in SP-treated cells during the G0-G2 transition deserves deeper investigation. For instance, AURKA has been shown to be essential for cilia disassembly in RPE cells (Pugacheva et al., Cell, 2007) and is also implicated in Golgi fragmentation (Persico et al., MBoC, 2010). Given this, could your pipeline be used to analyze AURKA localization relative to the centrosome or Golgi? Could you further link AURKA inhibition with Golgi fragmentation during G0-G1 transition? The mechanism presented here lacks clarity and detail.

Response:

We understand the reviewer's concern that some of the previously reported phenotypes, such as Taxol-induced Golgi fragmentation, may appear to be presented as novel findings in our previous manuscript. This was not our intention. These

phenotypes were included as validation examples to demonstrate the sensitivity and utility of our analytical pipeline. To avoid any misunderstanding, we have carefully revised the text (**highlighted with markers**) to clarify that these analyses were performed using well-established phenotypes for benchmarking purposes, not as new biological discoveries (e.g., lines 8–16 on page 9, lines 16–17 on page 10, and lines 1–5 on page 12 of the revised manuscript).

We also appreciate the excellent suggestion to further investigate the role of AURKA, and we have performed two additional analyses in response. First, we quantified AURKA accumulation at the centrosome and found that SP-treated cells exhibit a level of centrosomal AURKA comparable to that seen with AURKA knockdown (Fig 4A and Fig S5A of the revised manuscript). This result supports previous studies and suggests that centrosomal localization of AURKA is linked to the phenotypic changes observed upon cell cycle reentry in our study (additional section beginning on page 18, line 11 of the revised manuscript).

Second, we analyzed the spatial relationship between the centrosome and the Golgi apparatus (Fig 4B of the revised manuscript). Given that AURKA localizes to the centrosome, we interpret this as addressing the reviewer's suggestion to analyze AURKA localization relative to the centrosome or Golgi. Our analysis showed that the distance between the centrosome and Golgi does not markedly change under SP treatment or other experimental conditions (additional section beginning on page 18, line 11 of the revised manuscript). While this suggests that large-scale spatial rearrangement is unlikely to account for the observed phenotypes, we acknowledge that further investigation of functional links, such as material transport dynamics, between the Golgi and centrosome remains an important direction for future studies.

Comment:

- I have concerns regarding the ciliogenesis assays. The authors acknowledge that their observed ciliation rates are lower than previously reported (40% in their work vs. 70% in Wang & Brautigam, 2008, or 80% in Partisani et al., 2021). Is the cell line employed the best for this type of study?

Response:

We were also initially concerned about the relatively low ciliation rate observed in ARPE-19 cells. Although reported values vary considerably across studies, ARPE-19 cells are generally known to exhibit lower ciliation rates compared to other commonly used cell lines. A 40% ciliation rate after 48 hours of serum starvation, as

observed in our study, is therefore not unusually low in this context. We have added a clarification to this result in the revised manuscript (lines 23–25 on page 13 of the revised manuscript).

Due to the limited accessibility of foreign cell banks such as ATCC (technically feasible but requiring considerable effort and cost), we carefully selected a reliable source for obtaining the cell line and evaluated its suitability based on both literature evidence and extensive preliminary experiments. Based on these considerations, we believe the use of ARPE-19 cells is scientifically justified for the purposes of this study.

Comment:

Also, is confluency consistent across all experiments? For example, in Figure S7A, the top panels for serum +2 h and +18 h appear less confluent. Could variation in confluency affect ciliogenesis outcomes?

Response:

To minimize the potential influence of full confluency and polarization on cell cycle reentry, we intentionally seeded cells at relatively low density in this study. While minor variation across imaging fields is inevitable, we carefully selected 5–10 non-overlapping fields of view per condition to ensure representative sampling and confirmed the reproducibility of our observations. Therefore, we believe cell density was consistent enough across experiments to avoid introducing bias.

In response to the reviewer's comment, we have replaced the previously shown images with more representative examples (revised Fig S3A), and we have added a description regarding cell density control in the revised manuscript (lines 20–22 on page 13 of the revised manuscript).

Comment:

- Would using an additional cilia marker, such as Arl13B, yield similar results? Acetylation may be affected, but not the cilium itself.

Response:

As the reviewer rightly points out, deacetylation of tubulin can occur prior to ciliary disassembly. To address this concern, we used IFT88 as an additional cilia marker and confirmed its co-localization with acetylated tubulin under all experimental

conditions (revised Fig S4B). We did not observe any tubulin deacetylation that would affect the interpretation of our analyses in this study (from page 17, line 18 to page 18, line 2 of the revised manuscript).

Comment:

- siRNA experiments would benefit from data showing knockdown efficiency. A western blot or quantification panel should be included.

Response:

In response to the comment, we have included western blot data to show siRNA knockdown efficiency (revised Fig S5B).

Comment:

Minor comments:

- One focus of the study is Golgi unlinking during G2, yet the only synchronization method used is a double thymidine block. How do you confirm that cells are in G2, and particularly late G2? The inclusion of cell cycle markers would strengthen this claim. Since your method allows single-cell analysis, why not include a G2 marker (e.g., CENP-F) in an unsynchronized population to assess Golgi morphology more accurately?

Response:

In our preliminary experiments, we estimated the late G2 phase by identifying the time point just before a large proportion of cells entered mitosis following release from a double thymidine block, as described in the Methods section (corresponds to lines 15–17 on page 26 of the revised manuscript). While it is technically feasible to use cell cycle markers such as CENP-F, this would occupy one of the fluorescence channels, which is a practical limitation in our current imaging setup.

Moreover, more accurately identifying late G2 is inherently challenging, as it represents a short and somewhat ambiguous window within the cell cycle. We believe this is one of the key reasons for the variability observed in Golgi unlinking during this phase (both in previous studies and this study). Given the strengths of our single-cell approach, we expect that live-cell imaging, combined with retrospective cell cycle phase determination based on the timing of cell rounding at metaphase,

would offer a more precise way to identify late G2 without relying on additional markers. This strategy has been successfully used to define the onset of centriole duplication in previous studies (Takao et al., JCB, 2019 and Biol Open, 2019). With continued technical advances, we are eager to apply our method to live-cell imaging in future work, as we see great potential for deeper insights through this approach.

Comment:

- There appear to be duplicated images: Figures 2B and S4B, as well as Figures 4C and S9B, show identical representative fields. Please revise accordingly.

Response:

Our intention was to show individual channels in the supplemental figures and to include selected merged images from the same fields in the main figures for simplicity and clarity. While this was noted in the original figure legends, we recognize that it may have caused unnecessary confusion. To address this, we have reorganized the figures and removed all duplicated images in the revised manuscript.

Comment:

- In the ciliogenesis and G0-G2 transition figures, cilia are difficult to visualize. Separating the acetylated tubulin channel may improve clarity.

Response:

We have reviewed the figures throughout the manuscript and edited some of them to improve clarity, including adjustments to cropping to make key details such as cilia more visible. Due to space constraints, further layout optimization is challenging; however, we are happy to consider additional improvements if the current presentation remains unclear.

Comment:

- Depletion of JNK1 does not appear to affect the G0-G2 transition, while JNK2 and AURKA knockdowns phenocopy the SP inhibitor. Would you expect additive or synergistic effects from the co-depletion of JNK2 and AURKA?

Response:

We performed a double knockdown of JNK2 and AURKA (Fig S4A in the revised manuscript) and did not observe any synergistic effect. In addition, given that the phenotypes resulting from each single knockdown are not entirely identical (Fig 3D in the revised manuscript), we consider that JNK2 and AURKA function independently, at least in the context of G0/G1 transition. We have added a statement to this result in the revised manuscript (lines 9–17 on page 17). Nevertheless, the signaling pathways involving these factors are complex, and we plan to further investigate their potential interactions in future studies.

Reviewer #3:

Comment:

The authors present a study that attempts to use image-based morphological profiling to study the functions of the Golgi and cilia during the cell cycle. However, the work is largely descriptive, relying on morphometric data from low-resolution optical images of cells subjected to treatments such as small molecule and serum withdrawal. The manuscript suffers from several significant issues:

1) The conclusions are unconvincing, as the reported morphological phenotypes are, at best, only tangentially related to the underlying molecular and cellular mechanisms governing Golgi and cilia.

Response:

As this work was submitted as a Tools article mainly focusing on methodology, we acknowledge to some extent the reviewer's concern regarding the limited insight into underlying biological mechanisms. In the revised manuscript, we have added several new analyses, such as the quantification of AURKA accumulation at the centrosome (Fig 4A) and the spatial relationship between the centrosome and Golgi (Fig 4B), to strengthen the biological relevance of our findings (additional section beginning on page 18, line 11 of the revised manuscript). We believe these additions enhance the manuscript's biological insight to a level that is appropriate and sufficient for a JCB Tools article.

Comment:

2) While some morphological changes are detected through quantitative profiling, their functional significance remains unclear and is not further experimentally explored.

Response:

The analyses of mitosis (Fig 1 in the revised manuscript) and late G2 (Fig 2B in the revised manuscript) were primarily designed to validate the performance of our method using well-established phenotypes. We have revised the manuscript to clarify this point (highlighted with markers). Given that the biological significance of these phenotypes has already been extensively characterized in previous studies, we chose not to elaborate further to avoid redundancy. In contrast, for the G0/G1 transition, we believe that the additional analyses included in the revised manuscript provide deeper mechanistic insights and strengthen the biological relevance of our findings.

Comment:

3) Many hypotheses and claims lack experimental support. For instance, there is no direct evidence provided to substantiate the claim of Golgi unlinking.

Response:

If this comment refers to Golgi unlinking during late G2 (Fig S2 in the revised manuscript), we would like to clarify that this phenomenon has already been well described previously, as cited in the main text. If the concern instead relates to Golgi fragmentation during the G0/G1 transition, we believe that the additional analyses included in the revised manuscript, as outlined above (e.g., Fig 4A and B of the revised manuscript), now provide sufficient experimental support for our conclusions, meeting the standards expected of a Tools article.

Comment:

Overall, the study does not provide novel mechanistic insights into Golgi and cilium biology and is unlikely to make a significant impact on the broader field of cell biology. Its relevance may be limited to a niche audience focused on high-throughput morphological screening.

Response:

We believe that one reason high-throughput morphological screening is perceived as "niche" is the high barrier to entry in terms of cost, accessibility, and technical expertise. One of our goals is to help lower that barrier and make such approaches more broadly available. If our phenotypic profiling method can contribute to opening up high-content, high-throughput image analysis to a wider range of researchers, we would consider that a meaningful impact and a core achievement of this work.

September 15, 2025

Re: JCB manuscript #202503083R-A

Daisuke Takao
Huazhong Agricultural University

Dear Dr. Takao,

Thank you for submitting your revised manuscript entitled "High-Content Phenotyping Reveals Golgi Dynamics and Their Role in Cell Cycle Regulation." The manuscript has been seen by the original reviewers whose full comments are appended below. While the reviewers continue to be overall positive about the work in terms of its suitability for JCB, some important issues remain.

Our general policy is that papers are considered through only one revision cycle; however, in this case we are open to one additional short round of revision. Please note that we will expect to make a final decision without additional reviewer input upon resubmission.

You will see that a major concern is lack of comparison with existing methods, which is one of the main requirements for a JCB Tools paper. The final revision should include a clear example of an advantage over already available tools. We recognize that every tool comes with its own benefits and limitations, so we are not asking for a comprehensive comparison of multiple benchmarks among variable tools but rather for a highlight of a particular feature such as ease of use, computational advantages, etc. Please also add quantification of depletion efficiencies and address the other comments as needed with text changes.

Please submit the final revision within one month, along with a cover letter that includes a point by point response to the remaining reviewer comments.

Thank you for this interesting contribution to Journal of Cell Biology. You can contact me or the scientific editor listed below at the journal office with any questions at cellbio@rockefeller.edu.

Sincerely,

Alexey Khodjakov, PhD
Monitoring Editor
Journal of Cell Biology

Dan Simon, PhD
Scientific Editor
Journal of Cell Biology

Reviewer #1 (Comments to the Authors (Required)):

All concerns of this reviewer and other reviews' critiques have been addressed satisfactorily. The paper will be a valuable resource for the scientific community and is recommended for publication in the "Tools" category.

Reviewer #2 (Comments to the Authors (Required)):

Cao et al. addressed most of the prior concerns by adding essential controls, conducting additional experiments, and clarifying that the work's main aim is to validate a single-cell phenotypic-profiling pipeline rather than to report new phenotypes. The accompanying database, code, and README are clearly documented and easy to deploy.

The manuscript presents a useful tool for organelle-morphology analysis at single-cell resolution. That said, related tools such as SPACE, CellCognition Explorer, and ilastik, cover similar ground. The current version does not benchmark against these platforms, yet it claims improvement/refinement (page 20, lines 8-10). Absent head-to-head comparisons on shared datasets with standard metrics, the extent of improvement is unclear. Overall, the paper is stronger after revision, although the novelty of biological insight contribution is still limited.

I have noticed that the authors now include depletion efficiency by western blot (Fig. S5B). It would also be interesting if they

could quantify depletion. These experiments should also be described in the Materials and Methods section.

Reviewer #3 (Comments to the Authors (Required)):

The authors' revision is minimal and does not directly address my previous concerns.

While the morphological changes of cells can be described quantitatively, linking these changes to underlying molecular and cellular mechanisms is far more challenging, yet essential for developing a successful and useful tool. The manuscript presents a tool for calculating morphological descriptors; however, it fails to connect those descriptors to mechanisms.

1. The tool is not novel, as most quantifications described can be achieved by conventional methods. See below for a few examples.

2. The manuscript does not demonstrate how the tool reveals new molecular or cellular insights, particularly in Golgi and cilia biology. If the tool is indeed useful, the authors, being most familiar with it, should be able to provide such demonstrations.

Specific concerns:

1. Packed organization of the Golgi (Fig. 1B, Fig. S1B)

The molecular and cellular basis for the "packed organization" remains unclear, so even accurate quantification has limited interpretive value.

2. Role of GRASPs

a. The notion that GRASPs are essential for Golgi organization is controversial. Recent studies using acute degradation or knockout (e.g., PMID: 33301566) show that acute depletion of GRASP55 or GRASP65 does not affect Golgi ribbon structure.

b. Without independent verification of how the JNK2 inhibitor SP600125 affects Golgi stack linkage, the authors' study lacks a solid mechanistic basis.

3. Spindle pole detection

The use of α -tubulin fluorescence peaks to detect spindle poles is well established in the field and not novel. The extensive text devoted to this in the section "The image-based cell phenotype profiling identified features related to morphological defects in mitotic cells" is therefore unnecessary.

4. Golgi fragmentation

a) Changes in Golgi fragmentation can be detected using standard methods such as intensity segmentation followed by particle analysis.

b) Apparent fragmentation in low-resolution images may be misleading, as multiple Golgi stacks could remain connected by tubules that are not detected by low resolution and low sensitivity microscopy.

c) Imaging membrane linkages between Golgi stacks is technically challenging. The idea that the Golgi complex unlinks or fragments at G2/M is not a consensus view. Instead of adopting this idea, the authors should independently verify this claim rather than relying solely on previous publications.

Response to reviewer comments

We thank the reviewers and editors for their constructive feedback and valuable guidance. In direct response to the remaining concerns, we have made the following major revisions:

1. To highlight the ease of use of our tool, we added a description of its processing speed (revised manuscript, page 8, lines 3–15). The analysis can be readily performed on a standard office computer without requiring a special coding environment. For example, if microscopy images are already available, data comparable to those shown in Fig 1E–J can be obtained within half a day. We believe that making image analysis more accessible to a broader community of life scientists is one of the key advantages of our approach.

2. For the Western blot analysis of knockdown efficiency (Fig S5B), we added quantification data together with details of the quantification method (revised manuscript, page 26, lines 19–24, and legend of Fig S5B). In addition, we revised the Materials and Methods to describe antibodies, chemicals, siRNAs, and cell lines in text form rather than as bulleted lists.

Major revisions in the manuscript are **highlighted with markers**. In addition, we have prepared the code and dataset shared during the review process for public release on Dryad, and the link has been added to the Data and Code Availability section.

We sincerely thank the reviewers for helping us improve and refine the manuscript. Below, we provide a detailed point-by-point response to the individual comments.

Reviewer #1

Comment:

All concerns of this reviewer and other reviews' critiques have been addressed satisfactorily. The paper will be a valuable resource for the scientific community and is recommended for publication in the "Tools" category.

Response:

We sincerely thank the reviewer for the positive assessment and supportive comments. We are encouraged by the recognition of our work as a valuable

resource, and we hope that this tool will indeed benefit many researchers in the field.

Reviewer #2

Comment:

Cao et al. addressed most of the prior concerns by adding essential controls, conducting additional experiments, and clarifying that the work's main aim is to validate a single-cell phenotypic-profiling pipeline rather than to report new phenotypes. The accompanying database, code, and README are clearly documented and easy to deploy.

Response:

We sincerely appreciate the reviewer's constructive suggestions. It is our hope that many researchers will make use of this code and that imaging will become an even more accessible and widely adopted tool in future studies.

Comment:

The manuscript presents a useful tool for organelle-morphology analysis at single-cell resolution. That said, related tools such as SPACe, CellCognition Explorer, and ilastik, cover similar ground. The current version does not benchmark against these platforms, yet it claims improvement/refinement (page 20, lines 8-10). Absent head-to-head comparisons on shared datasets with standard metrics, the extent of improvement is unclear. Overall, the paper is stronger after revision, although the novelty of biological insight contribution is still limited.

Response:

We appreciate the reviewer's thoughtful comments and recognize the importance of benchmarking against existing tools. Our main emphasis in this work is on the simplicity and versatility of our pipeline, which makes it challenging to apply standard quantitative benchmarks such as those used for segmentation algorithms. Instead, we demonstrated its utility through practical analysis examples and by providing an accompanying database. As an additional indicator of accessibility, we included

processing time data and related descriptions in the revised manuscript (page 8, lines 3–15), highlighting that the code can be executed without high-performance computing resources. Additionally, our approach emphasizes the extraction of interpretable features, which provides advantages for hypothesis generation and validation compared with other classification methods, particularly deep learning-based approaches. Furthermore, we are pursuing independent projects to gain deeper biological insights and look forward to reporting these findings in future studies.

Comment:

I have noticed that the authors now include depletion efficiency by western blot (Fig. S5B). It would also be interesting if they could quantify depletion. These experiments should also be described in the Materials and Methods section.

Response:

We are grateful for this constructive suggestion. We have added the corresponding quantitative data, together with a description of the related methods (revised manuscript, page 26, lines 19–24, and legend of Fig S5B).

Reviewer #3

Comment:

The authors' revision is minimal and does not directly address my previous concerns.

While the morphological changes of cells can be described quantitatively, linking these changes to underlying molecular and cellular mechanisms is far more challenging, yet essential for developing a successful and useful tool. The manuscript presents a tool for calculating morphological descriptors; however, it fails to connect those descriptors to mechanisms.

Response:

We thank the reviewer for the continued engagement and for highlighting this important point. Our revision incorporated several new experiments and analyses

that we believe address many of the issues raised in the first review. For points where the reviewer's concerns remain, we are eager to engage in a constructive discussion. While the focus of some concerns in this round differs somewhat from those in the original review, we appreciate the reviewer's continued input and address these points explicitly in the responses below.

Comment:

1. The tool is not novel, as most quantifications described can be achieved by conventional methods. See below for a few examples.

Response:

We respectfully note that while the individual algorithms are established, the strength of our work lies in providing an integrated and streamlined pipeline that is immediately usable without specialized expertise. Indeed, even undergraduate students in our laboratory with no prior coding experience were able to apply the pipeline independently within a few weeks. This level of accessibility is not typical of existing approaches and represents a genuine advance in usability. Notably, the utility of the tool was acknowledged across reviewers in the first round of review. Our additional revisions, including explicit demonstrations of accessibility and processing speed (page 8, lines 3–15), further highlight this unique advantage.

Comment:

2. The manuscript does not demonstrate how the tool reveals new molecular or cellular insights, particularly in Golgi and cilia biology. If the tool is indeed useful, the authors, being most familiar with it, should be able to provide such demonstrations.

Response:

We greatly appreciate this important perspective. We agree that linking morphological features to molecular and cellular mechanisms is a crucial next step. In the present study, our primary aim was to establish and validate a systematic phenotypic profiling platform, rather than to provide a comprehensive mechanistic analysis. To illustrate the potential of this tool, we included examples in Golgi and cilia biology that already suggest heterogeneity not readily captured by qualitative observation alone. We view these findings as proof-of-principle demonstrations showing how our approach can highlight biologically meaningful variation and

generate hypotheses for future mechanistic work. Indeed, we are actively pursuing such studies, and we anticipate that combining this platform with targeted molecular perturbations will yield deeper insights.

Comment:

Specific concerns:

1. Packed organization of the Golgi (Fig. 1B, Fig. S1B)

The molecular and cellular basis for the "packed organization" remains unclear, so even accurate quantification has limited interpretive value.

Response:

We agree that the molecular basis of the "packed organization" remains to be clarified. However, a quantitative definition of such morphologies is essential for conducting subsequent mechanistic investigations. Our tool is not intended as a stand-alone solution, but rather as a framework to be combined with molecular and cell biological approaches, such as functional perturbation experiments. By enabling comprehensive single-cell phenotypic profiling, it provides a robust foundation for advancing our understanding of the molecular and cellular determinants of these phenomena.

Comment:

2. Role of GRASPs

a. The notion that GRASPs are essential for Golgi organization is controversial. Recent studies using acute degradation or knockout (e.g., PMID: 33301566) show that acute depletion of GRASP55 or GRASP65 does not affect Golgi ribbon structure.

Response:

We would like to clarify that our manuscript does not specifically discuss the function of GRASPs. While GRASP65 may be a potential target of JNK2, this is not a central focus of the current study. We appreciate that the role of GRASPs remains debated and regard it as an important question for future work, but it is not directly relevant to the validation of our phenotypic-profiling pipeline.

Comment:

b. Without independent verification of how the JNK2 inhibitor SP600125 affects Golgi stack linkage, the authors' study lacks a solid mechanistic basis.

Response:

We agree that the detailed mechanism by which SP600125 influences Golgi stack linkage remains to be clarified. Our knockdown experiments indicate that AURKA, rather than JNK2, plays a more prominent role during the G0/G1 transition, suggesting that the phenotype is unlikely to be solely attributable to JNK2 inhibition. While further analyses will be required to dissect the molecular pathway in detail, we consider such mechanistic studies beyond the scope of the current Tools manuscript. Our intent here is to demonstrate that the profiling pipeline can reliably capture relevant Golgi morphologies, thereby providing a useful platform to guide subsequent mechanistic investigations.

Comment:

3. Spindle pole detection

The use of α -tubulin fluorescence peaks to detect spindle poles is well established in the field and not novel. The extensive text devoted to this in the section "The image-based cell phenotype profiling identified features related to morphological defects in mitotic cells" is therefore unnecessary.

Response:

We respectfully disagree. The purpose of this section is not to claim novelty in spindle pole detection, but to transparently describe how we extracted and analyzed features. This information is important for readers, particularly those less experienced with image-based quantification, to understand how the pipeline functions in practice. We note that no other reviewer raised this concern, and we believe the description is an appropriate level of detail for reproducibility.

Comment:

4. Golgi fragmentation

a) Changes in Golgi fragmentation can be detected using standard methods such as intensity segmentation followed by particle analysis.

Response:

We agree that conventional methods, such as intensity-based segmentation followed by particle analysis, can be used to assess Golgi fragmentation. Our goal, however, is not to replace these algorithms but to provide an integrated and versatile platform that systematically handles multiple organelles within the same framework. Despite the availability of standard methods, qualitative scoring of Golgi morphology remains widespread in the field. By offering a low-barrier, easy-to-use tool, we aim to shift practice toward more reproducible quantification.

Comment:

b) Apparent fragmentation in low-resolution images may be misleading, as multiple Golgi stacks could remain connected by tubules that are not detected by low resolution and low sensitivity microscopy.

Response:

We recognize that the Golgi features discussed here are based on apparent morphology in relatively low-resolution images, and we have added a clarifying statement in the revised manuscript (pages 6, line 24 to page 7, line 2). While a more detailed description of Golgi structure would require additional analyses, examining these apparent morphologies can still provide valuable insights into cellular processes. Indeed, single-cell profiling revealed distinct subpopulations of cells exhibiting multiple morphological features under specific experimental conditions (Fig 2C). These differences are unlikely to be solely attributable to limitations in detection sensitivity or accuracy, but instead likely reflect biologically relevant changes in Golgi morphology. Thus, while our approach does not replace ultrastructural analysis, it provides an efficient first layer of quantitative information that can guide and prioritize subsequent mechanistic studies.

Comment:

c) Imaging membrane linkages between Golgi stacks is technically challenging. The idea that the Golgi complex unlinks or fragments at G2/M is not a consensus view. Instead of adopting this idea, the authors should independently verify this claim rather than relying solely on previous publications.

Response:

We agree that the idea of Golgi unlinking at G2/M has been discussed from multiple perspectives in the field and may not be considered a universal consensus. Our reference to “unlinking” follows prior definitions under specific conditions, and our intent is to demonstrate how our profiling pipeline can detect such morphologies. We have carefully evaluated the literature and used it as a basis for discussion, without attempting to resolve mechanistic controversies in this manuscript. Since the central focus of this study is the establishment of a profiling pipeline rather than detailed mechanistic analysis of the G2/M transition, we believe it is appropriate to illustrate its application without making strong claims about mechanistic consensus of Golgi unlinking at G2/M.

October 3, 2025

RE: JCB Manuscript #202503083RR

Daisuke Takao
Huazhong Agricultural University

Dear Dr. Takao,

Thank you for submitting your revised manuscript entitled "High-Content Phenotyping Reveals Golgi Dynamics and Their Role in Cell Cycle Regulation." We would be happy to publish your paper in JCB pending final revisions necessary to meet our formatting guidelines (see details below).

A. MANUSCRIPT ORGANIZATION AND FORMATTING:

1) Text limits: Character count for Tools is < 40,000, not including spaces. Count includes title page, abstract, introduction, results, discussion, and acknowledgments. Count does not include materials and methods, figure legends, references, tables, or supplemental legends.

2) Figure formatting: Tools may have up to 10 main text figures. Scale bars must be present on all microscopy images, including inset magnifications. Molecular weight or nucleic acid size markers must be included on all gel electrophoresis. Also, please avoid pairing red and green for images and graphs to ensure legibility for color-blind readers. If red and green are paired for images, please ensure that the particular red and green hues used in micrographs are distinctive with any of the colorblind types. If not, please modify colors accordingly or provide separate images of the individual channels.

3) Statistical analysis: Error bars on graphic representations of numerical data must be clearly described in the figure legend. The number of independent data points (n) represented in a graph must be indicated in the legend. Please indicate whether 'n' refers to technical or biological replicates (i.e. number of analyzed cells, samples or animals, number of independent experiments). If independent experiments with multiple biological replicates have been performed, we recommend using distribution-reproducibility SuperPlots (please see Lord et al., JCB 2020) to better display the distribution of the entire dataset, and report statistics (such as means, error bars, and P values) that address the reproducibility of the findings.

Statistical methods should be explained in full in the materials and methods. For figures presenting pooled data the statistical measure should be defined in the figure legends. Please also be sure to indicate the statistical tests used in each of your experiments (both in the figure legend itself and in a separate methods section) as well as the parameters of the test (for example, if you ran a t-test, please indicate if it was one- or two-sided, etc.). Also, if you used parametric tests, please indicate if the data distribution was tested for normality (and if so, how). If not, you must state something to the effect that "Data distribution was assumed to be normal but this was not formally tested."

4) Materials and methods: Should be comprehensive and not simply reference a previous publication for details on how an experiment was performed. Please provide full descriptions (at least in brief) in the text for readers who may not have access to referenced manuscripts. The text should not refer to methods "...as previously described." Please also describe the immunoblotting procedure including the type of membrane used, blocking reagent, and describe acquisition methods.

5) For all cell lines, vectors, strains, constructs/cDNAs, etc. - all genetic material: please include database / vendor ID (e.g. Addgene, ATCC, etc.) or if unavailable, please briefly describe their basic genetic features, even if described in other published work or gifted to you by other investigators (and provide references where appropriate). Please be sure to provide the sequences for all of your oligos: primers, si/shRNA, RNAi, gRNAs, etc. in the materials and methods. You must also indicate in the methods the source, species, and catalog numbers/vendor identifiers (where appropriate) for all of your antibodies, including secondary. If antibodies are not commercial, please add a reference citation if possible.

6) Microscope image acquisition: The following information must be provided about the acquisition and processing of images:

- a. Make and model of microscope
- b. Type, magnification, and numerical aperture of the objective lenses
- c. Temperature
- d. Imaging medium
- e. Fluorochromes

f. Camera make and model

g. Acquisition software

h. Any software used for image processing subsequent to data acquisition. Please include details and types of operations involved (e.g., type of deconvolution, 3D reconstitutions, surface or volume rendering, gamma adjustments, etc.).

7) References: There is no limit to the number of references cited in a manuscript. References should be cited parenthetically in the text by author and year of publication. Abbreviate the names of journals according to PubMed.

8) Supplemental materials: Tools may have up to 5 supplemental figures and 10 videos. Please also note that tables, like figures, should be provided as individual, editable files. A summary of all supplemental material should appear at the end of the Materials and methods section. Please include one brief sentence per item.

9) eTOC summary: A ~40-50 word summary that describes the context and significance of the findings for a general readership should be included on the title page. The statement should be written in the present tense and refer to the work in the third person. It should begin with "First author name(s) et al..." to match our preferred style.

10) Conflict of interest statement: JCB requires inclusion of a statement in the acknowledgements regarding competing financial interests. If no competing financial interests exist, please include the following statement: "The authors declare no competing financial interests." If competing interests are declared, please follow your statement of these competing interests with the following statement: "The authors declare no further competing financial interests."

11) A separate author contribution section is required following the Acknowledgments in all research manuscripts. All authors should be mentioned and designated by their first and middle initials and full surnames. We encourage use of the CRediT nomenclature (<https://casrai.org/credit/>).

12) ORCID IDs: ORCID IDs are unique identifiers allowing researchers to create a record of their various scholarly contributions in a single place. Please note that ORCID IDs are required for all authors. At resubmission of your final files, please be sure to provide your ORCID ID and those of all co-authors.

13) JCB requires authors to submit Source Data used to generate figures containing gels and Western blots with all revised manuscripts. This Source Data consists of fully uncropped and unprocessed images for each gel/blot displayed in the main and supplemental figures. For assays performed using capillary electrophoresis and/or immunoassay-based detection, authors should instead provide the electropherogram graph(s) for each experiment, plotting fluorescence/chemiluminescence intensity vs. molecular weight/size. Since your paper includes cropped gel and/or blot images, please be sure to provide one Source Data file for each figure gels, blots, and/or capillary electrophoresis assays along with your revised manuscript files. File names for Source Data figures should be alphanumeric without any spaces or special characters (i.e., SourceDataF#, where F# refers to the associated main figure number or SourceDataFS# for those associated with Supplementary figures). For traditional gels and blots, the lanes of the gels/blots should be labeled as they are in the associated figure, the place where cropping was applied should be marked (with a box), and molecular weight/size standards should be labeled wherever possible. For capillary electrophoresis assays, each trace in the graph should be color-coded and labeled to indicate which protein, gene, or sample is being measured (please try to avoid red/green combinations to accommodate our color-blind readers).

Source Data files will be directly linked to specific figures in the published article. Source Data Figures should be provided as individual PDF files (one file per figure). Authors should endeavor to retain a minimum resolution of 300 dpi or pixels per inch. Please review our instructions for export from Photoshop, Illustrator, and PowerPoint here: <https://rupress.org/jcb/pages/submission-guidelines#revised>

14) Journal of Cell Biology now requires a data availability statement for all research article submissions. These statements will be published in the article directly above the Acknowledgments. The statement should address all data underlying the research presented in the manuscript. Please visit the JCB instructions for authors for guidelines and examples of statements at (<https://rupress.org/jcb/pages/editorial-policies#data-availability-statement>).

B. FINAL FILES:

-- Cover images: If you have any striking images related to this story, we would be happy to consider them for inclusion on the journal cover. Submitted images may also be chosen for highlighting on the journal table of contents or JCB homepage carousel.

Images should be uploaded as TIFF or EPS files and must be at least 300 dpi resolution.

****It is JCB policy that if requested, original data images must be made available to the editors. Failure to provide original images upon request will result in unavoidable delays in publication. Please ensure that you have access to all original data images prior to final submission.****

****The license to publish form must be signed before your manuscript can be sent to production. A link to the electronic license to publish form will be sent to the corresponding author only. Please take a moment to check your funder requirements before choosing the appropriate license.****

Thank you for your attention to these final processing requirements. Please revise and format the manuscript and upload materials within 7 days. If you need an extension for whatever reason, please let us know and we can work with you to determine a suitable revision period.

Thank you for this interesting contribution, we look forward to publishing your paper in Journal of Cell Biology.

Sincerely,

Alexey Khodjakov, PhD
Monitoring Editor
Journal of Cell Biology

Dan Simon, PhD
Scientific Editor
Journal of Cell Biology